# The coincidence of ecological opportunity with hybridization explains rapid adaptive radiation in Lake Mweru cichlid fishes

Joana I. Meier [1,2,3,4], Rike B. Stelkens [1,2,5], Domino A. Joyce [6], Salome Mwaiko [1,2], Numel Phiri[7], Ulrich K. Schliewen[8], Oliver M. Selz [1,2], Catherine E. Wagner [1,2,9], Cyprian Katongo[7] & Ole Seehausen [1,2]*

The process of adaptive radiation was classically hypothesized to require isolation of a lineage from its source (no gene flow) and from related species (no competition). Alternatively, hybridization between species may generate genetic variation that facilitates adaptive radiation. Here we study haplochromine cichlid assemblages in two African Great Lakes to test these hypotheses. Greater biotic isolation (fewer lineages) predicts fewer constraints by competition and hence more ecological opportunity in Lake Bangweulu, whereas opportunity for hybridization predicts increased genetic potential in Lake Mweru. In Lake Bangweulu, we find no evidence for hybridization but also no adaptive radiation. We show that the Bangweulu lineages also colonized Lake Mweru, where they hybridized with Congolese lineages and then underwent multiple adaptive radiations that are strikingly complementary in ecology and morphology. Our data suggest that the presence of several related lineages does not necessarily prevent adaptive radiation, although it constrains the trajectories of morphological diversification. It might instead facilitate adaptive radiation when hybridization generates genetic variation, without which radiation may start much later, progress more slowly or never occur.

[1] Division of Aquatic Ecology & Evolution, Institute of Ecology and Evolution, University of Bern, Baltzerstr. 6, CH-3012 Bern, Switzerland. [2] Department of Fish Ecology and Evolution, Centre of Ecology, Evolution and Biogeochemistry (CEEB), Eawag Swiss Federal Institute of Aquatic Science and Technology, Seestrasse 79, CH-6047 Kastanienbaum, Switzerland. [3] Department of Zoology, University of Cambridge, Downing Street, Cambridge CB2 3EJ, UK. [4] St John's College, University of Cambridge, St John's Street, Cambridge CB2 1TP, UK. [5] Division of Population Genetics, Department of Zoology, Stockholm University, Svante Arrheniusväg 18 B, 106 91 Stockholm, Sweden. [6] Evolutionary and Ecological Genomics Group, Department of Biological and Marine Sciences, University of Hull, Hull HU6 7RX, UK. [7] Department of Biological Sciences, University of Zambia, Lusaka, Zambia. [8] SNSB-Bavarian State Collection of Zoology, Münchhausenstrasse 21, 81247 Munich, Germany. [9] Biodiversity Institute and Department of Botany, University of Wyoming, Laramie, WY 82071, USA. *email: ole.seehausen@eawag.ch

The role of interactions between species in facilitating or constraining lineage diversification is a central question in evolutionary biology. Adaptive radiations are classically thought to happen most often in isolated settings where colonizations are few and far apart in time, such that early colonists experience little competition with similar species[1,2] ('isolation hypothesis'). On the other hand, colonization by multiple closely related lineages may allow for hybridization between them, which can increase the evolutionary potential for diversification[3–7] ('hybridization hypothesis'). In addition, the presence of distantly related lineages that are not competing heavily for resources may facilitate diversification in any or several of these lineages by adding to the ruggedness of the adaptive landscape, such as when predators cause divergent selection on prey, or speciation in prey causes divergent selection on predators[8,9]. Divergent lineages may also provide ecological opportunities such as opportunities for commensalism, mutualism, parasitism or predation[9–11]. Here, we test if co-occurrence of multiple lineages facilitated or impeded adaptive radiation using African cichlid fishes.

Cichlid fishes have radiated in more than 30 lakes in Africa[12], and this includes some of the most spectacular animal radiations in terms of diversity of ecological and mating traits and the speed of evolution[13–17]. Some lake cichlid assemblages are mainly derived from a single lineage, or from two lineages that merged into a hybrid lineage (e.g. Lake Victoria region radiations), others are composed of multiple radiating lineages (e.g. Lake Tanganyika), and again others of multiple species that immigrated without any radiation[12,18]. In some cases, competition among the colonizing lineages may have constrained radiations[19], whereas in other cases co-occurrence of closely related lineages may have facilitated adaptive radiation through introgressive hybridization enriching the gene pool of the progenitor of the radiating lineage[4,7,20–23]. Indeed, a hybrid origin has recently been demonstrated for two of the three largest African cichlid radiations, those of Lakes Victoria and Tanganyika[4,22], and also for some of the large clades within the Malawi and the Tanganyika radiations[24–27].

Two East-African lakes, Lakes Mweru and Bangweulu, were previously not known to host any cichlid radiations despite the presence of multiple cichlid lineages[28] and favourable conditions for radiation (e.g. large lake size, low latitude, mating system with sexual selection and presence of sexual dichromatism[12]). The two lakes differ somewhat in their dimensions. Lake Mweru has a surface area of 5100 km$^2$ [29], with a mean depth of 7.5 m and a maximum depth of 27 m[30]. The surface area of Lake Bangweulu is subject to strong seasonal changes and is reported as 2500–5000 km$^2$ [29,31,32], with a mean depth of 4.7 m and a maximum depth of 10.4 m[33] (Supplementary Note 1). Previous work shows that for African lakes in general, lake depth is a predictive factor for adaptive radiation in cichlids, along with sexual dichromatism and energy measured as solar radiation[12]. However, the predictive power of lake depth on adaptive radiation strongly emerges only when lakes deeper than 100 m are included in the analysis, all of which host adaptive radiations. The scale at which depth is predictive of adaptive radiation vastly exceeds the difference in depth between Lakes Mweru and Bangweulu. Of the lakes with depths similar to Lakes Mweru and Bangweulu, some host cichlid adaptive radiations, whereas others show no in situ speciation at all (Supplementary Fig. 1). Lake surface area, on the other hand, predicts the species richness of cichlid assemblages but does not predict whether or not a radiation occurs[12]. Within the range of lake sizes samples, radiations are equally likely to occur in lakes with small surface area, but they attain lower species richness than large lakes[12,18]. The slightly smaller surface area of Lake Bangweulu would make us expect slightly lower species richness but is not expected to constrain the occurrence of adaptive radiation.

On the other hand, the two lakes differ strongly in their geological history. Proto-Lake Bangweulu formed in the Pleistocene in the Zambezi catchment and was much larger than the current lake[34,35]. Lake Bangweulu remained part of the Zambezi drainage for most of its geological history. Lake Mweru likely formed in the late Pleistocene[34] and belongs to the Upper Congo catchment. About 1 Mya, the outflow of Lake Bangweulu was captured by the Luapula diverting its waters towards Lake Mweru[35,36]. Zambezian cichlid (and other fish) lineages from Lake Bangweulu subsequently colonized Lake Mweru downstream, but waterfalls and extensive rapids make upstream movement of fish from Lake Mweru to Lake Bangweulu very difficult[28]. Cichlids in Lake Bangweulu have thus evolved in isolation from lineages elsewhere much more than those in Lake Mweru. Indeed, the fish fauna of Lake Mweru is composed of Congolese and Zambezian lineages, whereas Lake Bangweulu hosts almost exclusively Zambezian lineages (Fig. 1). The only cichlid exception is *Tylochromis bangwelensis*, a large non-haplochromine cichlid of Congolese ancestry with likely strong swimming capacities that seems to have colonized Lake Bangweulu upstream from Lake Mweru.

Here, we report an integrative investigation of the cichlid assemblages in both lakes, using phenotypic, phylogenomic and population genomic analyses to test predictions of the isolation and the hybridization hypotheses for adaptive radiation. The isolation hypothesis predicts reduced ecological opportunity in Lake Mweru due to competition among the much larger number of lineages, whereas the hybridization hypothesis predicts increased genetic opportunity in Lake Mweru compared to Lake Bangweulu because of the presence of more related lineages that can hybridize. Through extensive sampling of these two lakes and examination of more than 1400 preserved fish, we confirm that radiations are absent in Lake Bangweulu, but in Lake Mweru, we find an impressive diversity of undescribed haplochromine cichlids. We identify more than 40 phenotypically distinct, putative species in Lake Mweru with up to 15 co-occurring at single collecting sites, varying in size, morphology, male breeding colouration, habitat use and feeding adaptations. With genome-wide sequence data, we show that they belong to multiple adaptive radiations that evolved largely contemporaneously. All of these radiations are derived from hybrid ancestry between Congolese and Zambezian lineages. In contrast, we find no evidence for hybridization among lineages in Lake Bangweulu where no radiations evolved. We also provide indirect evidence from patterns in ecology and morphospace occupation for the prediction that cichlid lineages diversifying in the same lake interact ecologically in ways consistent with competition as well as predator–prey interactions among lineages. Together, these mechanisms generate an assemblage of multiple radiations in Lake Mweru that is comparable to the mega-radiations of Victoria, Malawi and Edward in phenotypic and ecological diversity, whereas no radiation has occurred in Lake Bangweulu.

## Results

**Taxonomic diversity in Lake Bangweulu and Lake Mweru**. We caught and identified over 600 cichlids in Lake Bangweulu at eight sites, of which we preserved 404. We also caught and identified over 1200 cichlids in Lake Mweru at 12 sites and at six sites on the Kalungwishi and Luongo/Luapula Rivers that flow into Lake Mweru, of which we preserved 1066. Our samples from Lake Bangweulu were readily identified morphologically to ten described species[28]. These included six haplochromine cichlids, five of which are widespread in the Zambezi. The sixth species is restricted to Lakes Bangweulu and Mweru and the Luapula River

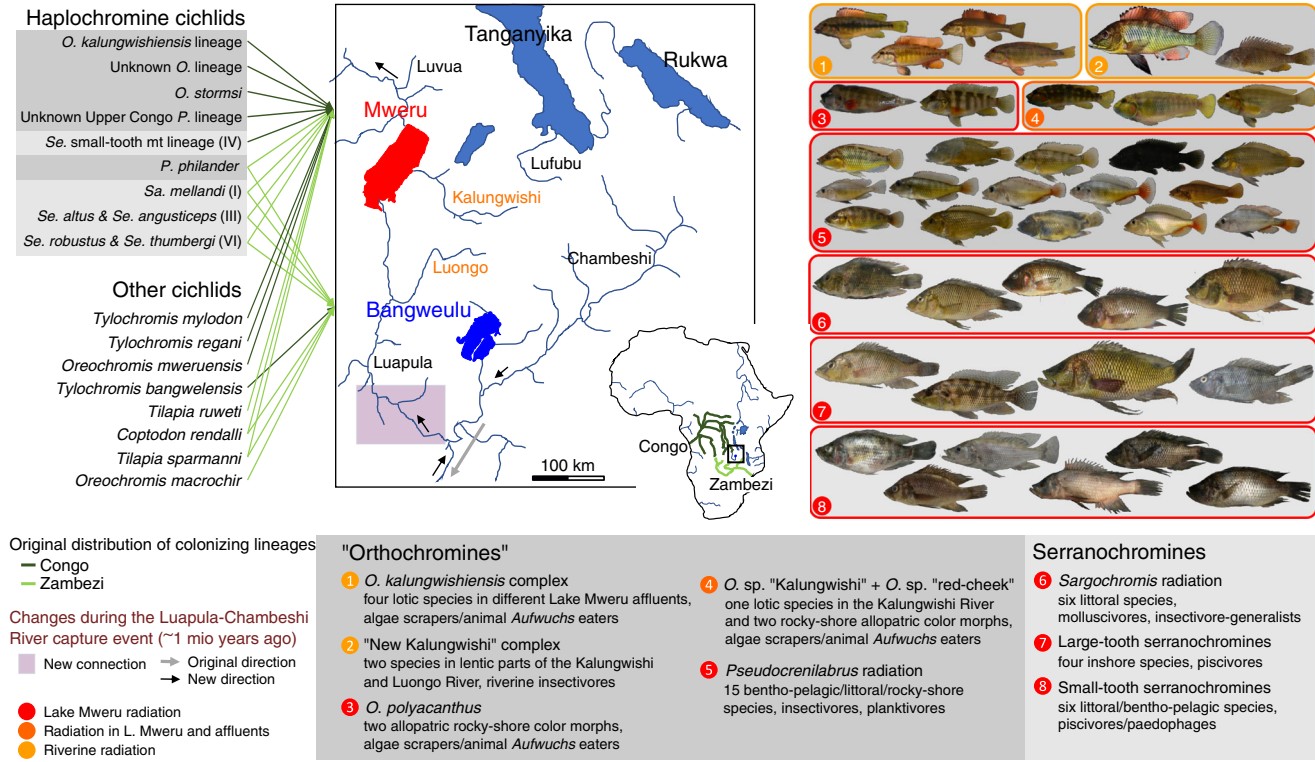

**Fig. 1** Geographic setting and cichlid diversity in Lakes Bangweulu and Mweru. Map of the Lake Mweru–Bangweulu region with colonizing lineages of haplochromine cichlids from the genera *Serranochromis* (*Se.*), *Sargochromis* (*Sa.*), *Pseudocrenilabrus* (*P.*) and *Orthochromis* (*O.*), and other cichlids. Lake Mweru has always been part of the Congo catchment (dark green in the inset map of Africa). During the Pleistocene, about 1 million years ago, the Luapula River captured the northeastern arm of the Zambezi catchment (light green), the Chambeshi–Bangweulu system. Lake Mweru was likely colonized by 11 species belonging to nine lineages of haplochromine cichlids. The fish photos illustrate the species that evolved from these lineages in radiations restricted to Lake Mweru (red frame) or with species in rivers flowing into Lake Mweru (orange frame). The number of species and overall ecology of the radiations are listed below. Lake Bangweulu was colonized by six haplochromine species belonging to four lineages, all of which are widespread in the Zambezi and none of them speciated in the lake. Mitochondrial haplotype clade numbers following Joyce et al.[42] are given in parentheses. Both lakes were colonized by non-haplochromine cichlids too (four species in Lake Bangweulu, six in Lake Mweru) but none of these diversified in the lakes and they are not part of this study. The photos were taken by Ole Seehausen, except for the "New Kalungwishi" individuals which were photographed by Hans van Heusden (first photo) and by Numel Phiri and Cyprian Katongo (second photo), and the four photos of the *O. kalungwishiensis* complex which were also taken by Hans van Heusden. The species names are given in Supplementary Fig. 2.

but is closely related to Zambezi taxa (Fig. 1, Supplementary Note 2). In contrast to the modest diversity in Lake Bangweulu, we encountered a spectacular diversity of haplochromine cichlids with mostly previously unknown species in Lake Mweru. In Lake Mweru and two inflowing rivers, the Kalungwishi and the Luongo, a tributary of the Luapula (Fig. 1), we recorded six non-haplochromine cichlids (all previously described) (Supplementary Figs. 2, 3). We also found each of the haplochromine species that are also represented in Lake Bangweulu (Supplementary Fig. 3). In addition, we discovered about 40 putative species of haplochromine cichlids previously unknown to science and endemic to Lake Mweru or its inflowing rivers (Supplementary Figs. 2, 4, Supplementary Note 3). Genomic and morphological data revealed that they evolved in four larger radiations, all confined to Lake Mweru (hereafter called lacustrine radiations, red circles in Fig. 1), and four smaller radiations with endemic species in the lake and inflowing rivers (Fig. 1). The radiations form part of two highly divergent groups, the serranochromines and a second group including the genera *Orthochromis* and *Pseudocrenilabrus* as well as some undescribed relatives, that we informally collectively refer to subsequently as "orthochromines" (Fig. 1). "Orthochromines" encompass four smaller radiations with species of the genus *Orthochromis* in the lake and adjacent rivers and a lacustrine *Pseudocrenilabrus* radiation, which is the largest of all

radiations in the Mweru system with 15 putative species, and the only lacustrine radiation of *Pseudocrenilabrus* known in all of Africa (Supplementary Figs. 2, 4, Supplementary Note 3). We (R.B.S. and O.S.) performed mate choice experiments in aquaria with three of these putative species and showed that they mate strongly species-assortatively[37]. We also bred these putative species in aquaria and found them to breed true. Serranochromines contain the three other lacustrine radiations comprising six *Sargochromis* putative species and two radiations in the genus *Serranochromis*. One *Serranochromis* radiation includes four mostly inshore predators with large teeth, which we named 'large-tooth serranochromines'. The second radiation, which we call 'small-tooth serranochromines', includes six littoral or bentho-pelagic putative species with noticeably smaller teeth (Supplementary Figs. 2, 4, Supplementary Note 3).

We identified individuals to putative species in the field and subsequently analysed the radiations in the Lake Mweru catchment morphologically and genetically (Supplementary Fig. 4). 'Species' as a factor explained highly significant amounts of variance on one or more morphological and also on one or more genomic PC axes within each lacustrine radiation except for the genomic PCA of the large-tooth serranochromines, but sample sizes of the latter were small (Supplementary Fig. 4, Supplementary Note 3).

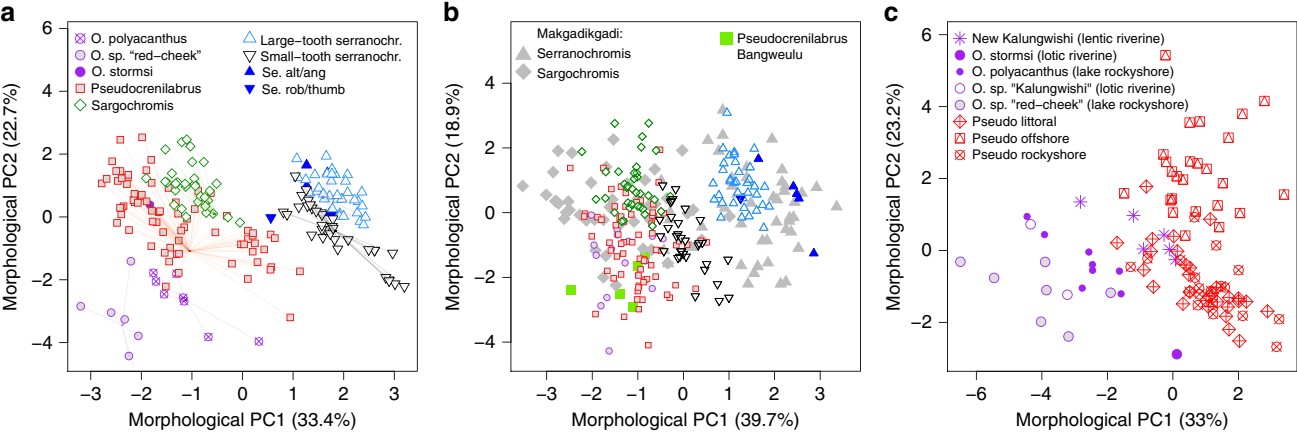

**Fig. 2** Morphological variation and morphospace partitioning between Lake Mweru radiations. **a** Morphological complementarity of the four intralacustrine cichlid radiations, the lacustrine members of the *Orthochromis* assemblage, the non-endemic *Serranochromis altus* and *Se. angusticeps* (Se. alt/ang), the nonendemic *Se. robustus* and *Se. thumbergi* (Se. rob/thumb) and *Orthochromis stormsi* (O. stormsi) in Lake Mweru. The first two principal components of all haplochromine cichlid species we found in Lake Mweru show nearly perfect complementarity in morphospace occupation among the radiations. Of the *Orthochromis* species, only the taxa occurring in Lake Mweru, *O. polyacanthus* and *O.* sp. "red-cheek" are shown, but not the seven riverine species of the Kalungwishi and Luongo Rivers. Sister taxa of each radiation are indicated with filled symbols. Their centroids or that of the radiations themselves (in the absence of sister taxa) are connected by thin lines to each phenotype of the corresponding radiation to visualize approximate trajectories of phenotypic divergence and diversification. The underlying data and the R script for all panels are provided at Zenodo with doi: 10.5281/zenodo.3435419. **b** Mweru taxa predicted onto morphospace of serranochromines of the radiation of ancient Lake Makgadikgadi. Makgadikgadi *Sargochromis* include *Chetia* and *Pharyngochromis* species which are nested in the genus *Sargochromis*[42]. In the presence of a diverse *Pseudocrenilabrus* radiation in Lake Mweru, the serranochromine radiations altogether are confined to a subset of the morphospace this lineage occupies elsewhere (such as in the Okavango region, the modern centre of Makgadikgadi-derived diversity). This is mostly due to much reduced morphological diversity in *Sargochromis* of Lake Mweru. Compared to *Pseudocrenilabrus* of Lake Bangweulu, the *Pseudocrenilabrus* radiation in Lake Mweru expanded into serranochromine morphospace. **c** In the presence of the *Pseudocrenilabrus* radiation in the lake, *Orthochromis* are confined to the epilithic algae and Aufwuchs scraper niche in the rocky wave washed littoral of the lake, an extreme feeding niche that the *Pseudocrenilabrus* radiation has not invaded. In the rivers, where *Pseudocrenilabrus* have not radiated and only the ancestral type *P. philander* is present, two species with partial *Orthochromis* and partial *Pseudocrenilabrus* ancestry evolved in lentic, i.e. lake-like, parts of the river. These "New Kalungwishi" species overlap in morphospace with littoral *Pseudocrenilabrus* from Lake Mweru.

**Mweru radiations are complementary in ecology and morphospace**. The radiations of Lake Mweru show remarkable complementarity with each other in morphology and ecology (Figs. 1, 2). Each radiation occupies different and unique sectors in morphospace that mirror trophic and habitat niche space (Fig. 1). We inferred the most likely trophic niche from dentition and morphology[38], and the habitat occupancy from presence/absence and abundance at our sampling sites. *Orthochromis* species in the lake (*O.* sp. "red-cheek" and *O. polyacanthus*) are confined to rocky shores, where they scrape algae and other *Aufwuchs* (periphyton and benthic invertebrates). *Orthochromis* in the rivers (*O.* sp. "Kalungwishi" and species of the *Orthochromis kalungwishiensis* complex) are confined to lotic (running water) habitats taking algae and insects from rocky substrate. The two species of the "New Kalungwishi" complex feed on insect larvae in lentic (lake-like) extensions of the Kalungwishi and the Luongo River. Species of the *Pseudocrenilabrus* radiation feed on insect larvae, and small benthic and planktonic crustaceans in a wide range of habitats. Members of the *Sargochromis* radiation are littoral molluscivores or feed on large insect larvae. Large-tooth and small-tooth serranochromines feed in several different habitats on large insect larvae and smaller fish, likely including the "orthochromine" species, their eggs and fry, and small cyprinid fish. Whereas for some of the genera, these trophic niches are typical across much of their African range, the largest Mweru radiation occurred in the genus *Pseudocrenilabrus* which otherwise contains exclusively trophically unspecialized microinvertivorous wetland species[39,40]. In Lake Mweru, however, they have diversified into all available habitats (littoral soft substrates, rocky shores and demersal offshore) and have adopted a variety of feeding styles and prey.

"Orthochromines" are generally smaller than serranochromines (Supplementary Fig. 5). We measured ecologically relevant morphological traits[41] to assess morphospace occupation as a proxy for niche partitioning between species and among radiations. In morphological PCA space of all Lake Mweru haplochromines combined, we observed nearly no overlap in morphospace among the lacustrine radiations (Fig. 2a). Investigating the trajectories of phenotypic diversification of each radiation compared to ancestor proxies, we identified patterns that are suggestive of limits to divergence and diversification imposed by the presence of the other sympatric radiations (Fig. 2a). In comparison to the radiation of serranochromines that arose in paleolake Makgadikgadi (many of which persist today in the Okavango region)[42], the serranochromines in Lake Mweru lack representatives in a large part of the Makgadikgadi/Okavango morphospace which in Mweru is occupied by "orthochromines" (Fig. 2b). In the Makgadikgadi/Okavango region, in contrast, the "orthochromines" are represented by a single wetland generalist, *Pseudocrenilabrus philander*. Moreover, patterns of diversity in *Orthochromis* and *Pseudocrenilabrus* in Lake Mweru are consistent with evolutionary effects of competition in sympatry: in Lake Mweru, these genera do not overlap in morphospace (Fig. 2a). While *Pseudocrenilabrus* have diversified into a wide range of insectivores and planktivores in many different habitats, *Orthochromis* in the lake are confined to shallow water rocky shore algae/*Aufwuchs* scraper niches. However, in the Kalungwishi River, where the only *Pseudocrenilabrus* is a wetland generalist, the "New Kalungwishi" lineage which mitochondrially belongs to *Orthochromis*, and is genomically admixed between *Orthochromis* and *Pseudocrenilabrus*, formed two endemic species that occupy lake-like habitats with

quiet water and soft bottoms associated with morphologies that converge in morphospace on the littoral members of the *Pseudocrenilabrus* radiation (Fig. 2c).

The complementarity in morphospace among the radiations is consistent with competition among lineages having constrained the adaptive radiations and having shaped niche partitioning among them in sympatry. At the same time, ecological interactions among the most distantly related and ecologically most distinct haplochromine lineages may have facilitated radiation in Lake Mweru. The *Pseudocrenilabrus* species invaded a variety of littoral, rocky-shore and offshore habitats and may have facilitated radiation of the two lineages of large piscivorous *Serranochromis* into these diverse habitats by serving as prey, a hypothesis that should be tested with ecological data in the future.

**All Lake Mweru radiations evolved rapidly and recently**. Next, we wanted to know if the Lake Mweru radiations evolved before or after the Chambeshi/Luapula River capture event and if the different radiations were of similar age. We found that within all four lacustrine radiations of Lake Mweru, mitochondrial branch lengths are short and mitochondrial lineage sorting between putative species is highly incomplete. We estimated the age of the population expansions associated with the radiations using BEAST v. 2.5.0[43] on subsets of the six most divergent mitochondrial haplotypes for each radiation, excluding individuals with cyto-nuclear mismatch among radiations. We added the closest relative from outside each radiation and some outgroups. Note that the deepest split in a haplotype clade may predate the onset of the corresponding species radiation, as some of the haplotype diversity may already have been present in the ancestral population. As the correct calibration of cichlid trees is controversial[44], we used four different calibration sets with priors on the age of the Lake Malawi cichlids and on the modern haplochromines (see Supplementary Table 1).

The 95% confidence intervals around the age of the radiations overlap broadly with each other and with the age estimate of the Lake Victoria Region Superflock of haplochromine cichlids, which is thought to be 100,000–200,000 years old[45–47] (Supplementary Fig. 6, Supplementary Note 4). This suggests that the radiations may have occurred largely contemporaneously and recently. Depending on the calibration set used, the deepest splits in the mitochondrial haplotype clades of *Pseudocrenilabrus*, the large-tooth serranochromines, the *Sargochromis* radiation, and the small-tooth serranochromines are dated to 0.27–0.35, 0.43–0.56, 0.27–0.94, 0.72–0.94 Mya, respectively. These age estimates may be overestimates due to the time-dependency of the molecular clock in the most recent 1 Mya[15,48–50]. The absence of mitochondrial haplotype sorting amongst putative species within any of the lake radiations (Supplementary Fig. 6, Supplementary Data 1) suggests that speciation must be considerably more recent than the split of the most divergent mitochondrial haplotypes in each radiation. Yet, the age estimates of the haplotype radiations are all younger than the Luapula-Chambeshi River capture event which is estimated to about 1 Mya[35,36]. The radiations both in Congolese and Zambezian lineages of Lake Mweru thus appear to have begun only after the Zambezian lineages could have arrived in Lake Mweru. Similarly, we estimated the crown age of the group including O. sp. "red-cheek" and O. sp. "Kalungwishi" as 0.36–0.47 Mya depending on the calibration set used. In contrast, the entirely riverine *O. kalungwishiensis* complex seems to be older (1.20–1.57 Mya) and likely predates the origin of the Mweru–Bangweulu connection. The species in this group are exclusively allopatrically distributed (different rivers) and qualify as a non-adaptive radiation rather than as an adaptive radiation[51,52].

Despite considerable overlap in the confidence intervals of all lake radiations, the mitochondrial clade of the small-tooth serranochromines might be slightly older than those of the three other lake radiations (the mean age estimate of the deepest split is roughly twice that in the other radiations). This deeper mitochondrial divergence in the small-tooth serranochromines may reflect larger ancestral haplotype diversity. This scenario is likely because the mitochondrial lineage of the small-tooth serranochromines does not have any close relatives outside the lake and may thus have been present in the Lake Mweru drainage long before the connection to the Bangweulu drainage became established, and possibly long before the beginning of adaptive radiation.

A single individual of the *Pseudocrenilabrus* radiation of Lake Mweru has a mitochondrial haplotype that groups it with *P. philander* of Lake Bangweulu and Lake Chilwa. This individual split from those *P. philander* 0.40–0.52 Mya (Supplementary Fig. 6), after the Luapula-Chambeshi capture event. This time also coincides with the estimated age of the beginning of the expansion of the majority mitochondrial clade in the *Pseudocrenilabrus* radiation, consistent with introgression of the Bangweulu/Chilwa mitochondrial haplotype before or around the time of the onset of the *Pseudocrenilabrus* radiation in Lake Mweru (see below).

**Hybrid origins of every Lake Mweru radiation**. Next, we wanted to understand how the radiations could all have emerged so rapidly within less than a million years in Lake Mweru, and why none evolved in Lake Bangweulu. For this, we tested the hypothesis that rapid radiations may have been preceded by introgressive hybridization between distinct colonizing lineages and that lineages that did not radiate had not received introgression.

Like the classical cichlid radiations in other African lakes, the individual radiations in Lake Mweru are largely monophyletic in the mitochondrial tree and fully monophyletic in the genome-wide nuclear tree (Supplementary Fig. 3). However, large cytonuclear discordance (Supplementary Fig. 3), rare cases of mitochondrial haplotype sharing across radiations (Supplementary Fig. 3) and genomic tests of hybridization (D statistics, f4 tests[53] and MixMapper[54], Supplementary Data 2–9) revealed evidence for rampant ancestral hybridization at or near the origins of every Lake Mweru radiation (Fig. 3, Supplementary Note 5). The hybridization tests were robust to the choice of outgroups (Supplementary Data 2–4, 7, 8). Genomic clustering analyses with ADMIXTURE[55] revealed no evidence for recent gene flow between radiations (Supplementary Fig. 7). In addition, signatures of excess allele sharing were consistent and homogeneous across species within each radiation (Supplementary Data 3, 7) suggesting that hybridization took place before the radiation processes started, consistent with the hypothesis that it could have fuelled diversification. Haplotype sharing patterns identified by fineRADstructure broadly support the absence of recent gene flow and the ancient hybridization events inferred with D statistics (Supplementary Figs. 8, 9). The admixture graph method MixMapper[56] confirmed the evidence for hybrid origins of the radiations and provided estimates for ancestry proportions (Supplementary Data 6, 9). All four lacustrine radiations in the melting pot Lake Mweru seem to be of hybrid origin between Zambezian and Congolese lineages. In addition, the two riverine species in the "New Kalungwishi" clade show strong signatures of hybrid ancestry between *Orthochromis* and *Pseudocrenilabrus* (Fig. 3, Supplementary Data 8). This suggests that introgression from *Pseudocrenilabrus* may have facilitated ecological niche expansion in a lineage that is *Orthochromis* in its mitochondrial

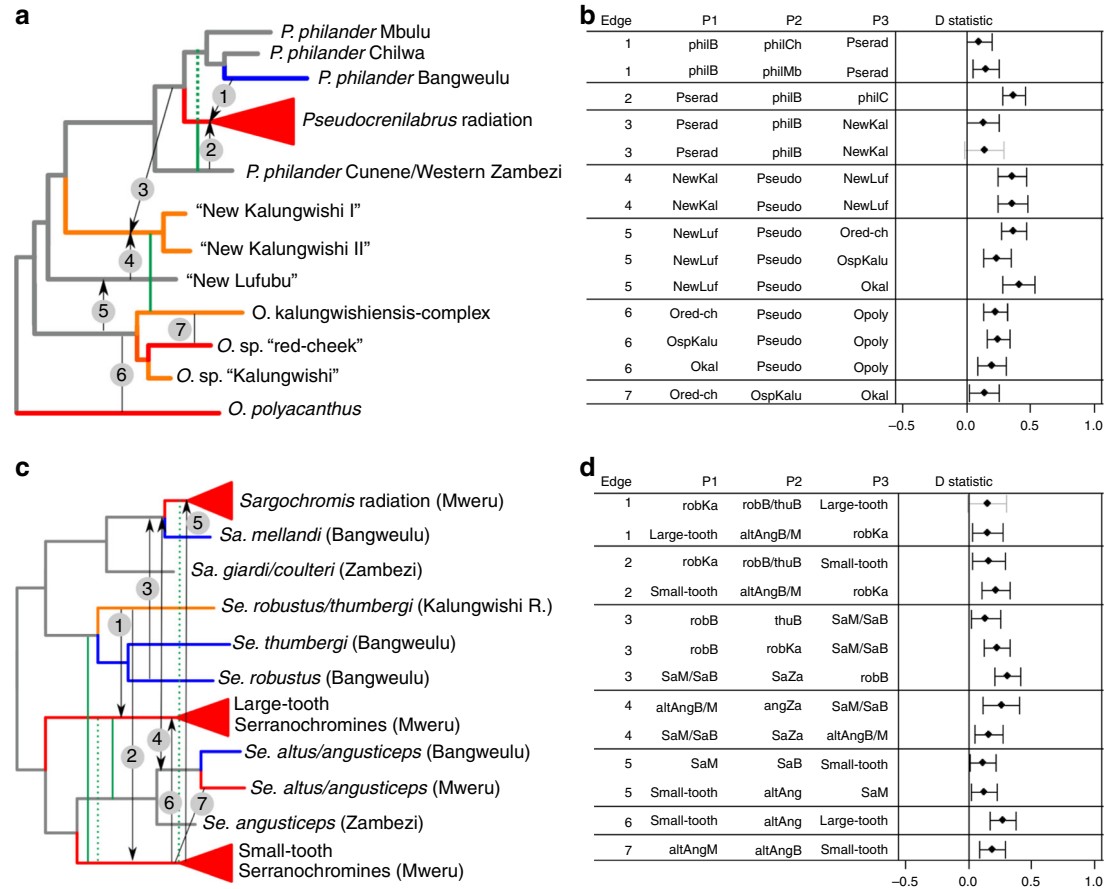

**Fig. 3** Cytonuclear discordance and tests of hybridization reveal reticulate ancestry of all Mweru radiations. Nuclear cartoon topology of the "orthochromines" (**a**) and serranochromines (**c**) with green vertical lines showing mitochondrial sister relationships that deviate from the nuclear tree (solid lines) and sharing of mitochondrial haplotypes (dashed lines). Lake Mweru taxa are highlighted with red edges and radiations are shown as triangles, whereas Lake Bangweulu taxa are shown with blue edges. Riverine taxa in the drainage system of Lake Mweru are shown with orange edges. Black vertical lines indicate evidence for hybridization from D statistics (numbered as shown in **b** and **d**) and other tests of hybridization. Where data allow, arrow heads indicate the putative direction of gene flow. **b** and **d** D statistic results supporting hybridization edges shown in **a** and **c** for "orthochromines" and serranochromines, respectively. Error bars indicate three standard deviations from the mean and are depicted in grey if overlapping with 0 (non-significant, |z| < 3). D statistics and z-scores are averaged across multiple tests with different outgroups. Tests for edges 4–6 in **b** show averages of tests using *Pseudocrenilabrus* from Lake Mweru, Bangweulu or from the Zambezi/Cunene as P2 as they were all very similar. Likewise, tests where two groups are given for P2 or P3 revealed very similar results for both groups and have thus been averaged. All D statistics used to compute the averages are provided as Supplementary Data 2–4 and 7–8. R scripts and the data underlying the figures are provided at Zenodo with https://doi.org/10.5281/zenodo.3435419.

lineage. Other riverine *Orthochromis* species also show evidence for past hybridization (Supplementary Data 8) but this did not seem to have spurred further diversification, probably due to lack of ecological opportunity in the lotic river habitat[17].

**Lack of radiations and of hybridization in Lake Bangweulu.** Whereas in Lake Mweru cichlids cytonuclear discordance at the base of each radiation is strong and genome-wide tests of hybridization support admixed ancestry of every radiation (Fig. 3, Supplementary Fig. 3), much less evidence for hybridization was detected in Lake Bangweulu (Fig. 4, Supplementary Data 10). No mismatches between phenotypic taxonomy and mitochondrial haplotype clade were observed among Lake Bangweulu taxa (Supplementary Fig. 3). The only hybridization signals involving Lake Bangweulu taxa are signals of excess allele sharing and cytonuclear discordance that affect *Sargochromis* of both Lake Mweru and Lake Bangweulu (Fig. 4): Compared to *Serranochromis robustus/thumbergi* of the Kalungwishi River, *Se. robustus* of Lake Bangweulu shows excess allele sharing with *Sargochromis* of both Lakes Mweru and Bangweulu (Fig. 4, Supplementary

Data 10). However, the facts that *Sargochromis* from both lakes show equally strong allele sharing with *Se. robustus* and that *Se. thumbergi* from Lake Bangweulu also shows slightly elevated allele sharing with both *Sargochromis* clades, suggest that introgression was likely from *Se. robustus* into the common ancestor of both *Sargochromis* clades. This likely reflects a hybridization event in the relatively distant past rather than hybridization restricted to Lake Bangweulu. None of the Lake Bangweulu taxa show any additional evidence for hybridization (Fig. 4, Supplementary Data 10).

The absence of hybridization signals in *Pseudocrenilabrus* from Lake Bangweulu is easily explained by the absence of close relatives in the catchment. However, with the one exception discussed above, we also found no evidence of hybridization among serranochromines, even though all species are sufficiently closely related to hybridize. This may be due to their long history of sympatric coexistence and the evolution of behavioural reproductive isolation. In contrast, in Lake Mweru, lineages came into contact that had previously evolved in allopatry, separated in different drainage systems, and thus may hybridize more readily. In line with this argument of larger potential for hybridization

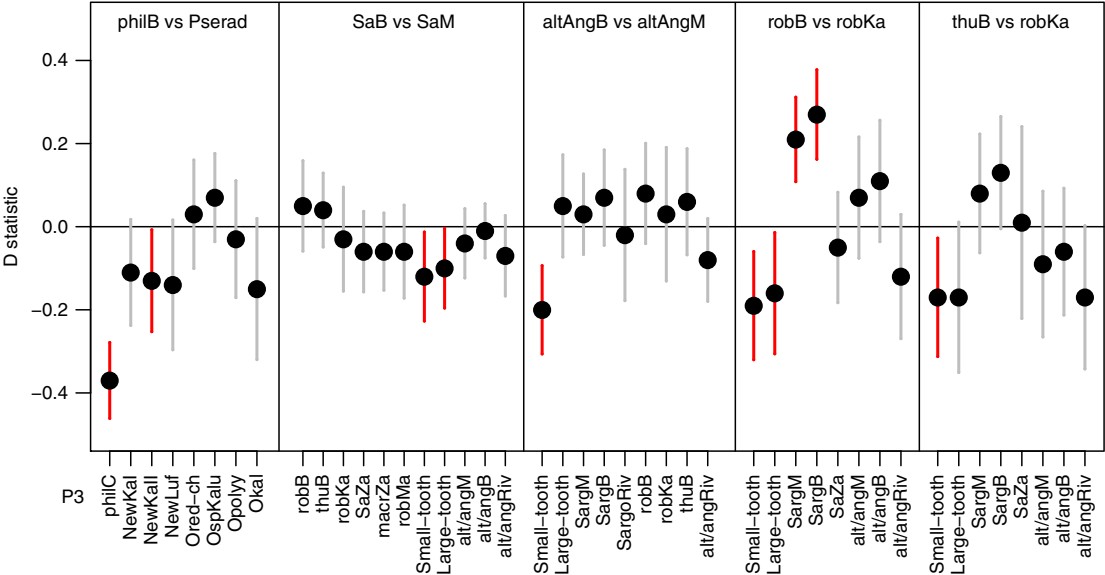

**Fig. 4** Lake Bangweulu taxa show no signatures of hybridization. Comparisons of Bangweulu taxa (as P1) with their closest relatives (sister taxon in Lake Mweru as P2) reveals no evidence for excess allele sharing between any Bangweulu taxon with any other taxon (P3) in our dataset, which would lead to positive D statistics. The only exception is *Se. robustus*, which shows excess allele sharing with *Sargochromis* both of Lake Mweru and of Lake Bangweulu. Given that the two *Sargochromis* lake clades do not differ in allele sharing with *Se. robustus* of Bangweulu (see first test of 'SaB vs SaM' in this figure) and that slight excess allele sharing is also observed in the closely related *Se. thumbergi* of Lake Bangweulu, gene flow likely occurred from *Se. robustus* Bangweulu into the common ancestor of both *Sargochromis* lake clades. Therefore, the onxly signature of hybridization involving a Bangweulu taxon is shared by both Bangweulu and Mweru taxa and may reflect hybridization in the distant past. Hence, there is no additional event of hybridization in Lake Bangweulu, whereas there is rampant evidence for hybridization among multiple different Lake Mweru taxa (see Fig. 3). Abbreviations and colour scheme follow those in Fig. 3. Exact values are given in Supplementary Data 10. The R script and the input file used to make this figure are provided at Zenodo with https://doi.org/10.5281/zenodo.3435419.

between allopatrically evolved lineages, *Pseudocrenilabrus* from Lake Mweru, when tested in mate choice experiments, showed no premating isolation against phenotypically similar *Pseudocrenilabrus* from Lake Bangweulu. However, they did show premating isolation against other sympatric *Pseudocrenilabrus* species from Lake Mweru despite much smaller genetic distance compared to the allopatric lineages tested[37].

## Discussion

Many adaptive radiations occurred in insular biota where colonizing lineages are few and their arrivals far apart in time and thus only early colonists find vacant niches that provide ample ecological opportunity for evolutionary diversification. This has led to the expectation that adaptive radiations most readily emerge from single lineages evolving in the absence of competing lineages[1]. However, the use of genome-scale analyses of adaptive radiations has revealed that many adaptive radiations are of hybrid ancestry either in parts or in their entirety[4,7,22,23,57–63]. In the presence of ecological opportunity, hybridization may facilitate and speed up adaptive radiation because of the rapid provisioning of large amounts of genetic variation with large effects on ecology, fitness and mating traits[3,5,6,64]. Such effects of hybridization may be maximized at intermediate genetic distance between lineages, as observed in hybrid speciation experiments[65], which likely also applies to adaptive radiations derived from hybrid ancestry[7]. This is because hybridization between closely related lineages does not generate much genetic novelty[66,67] but when lineages are too distantly related, intrinsic incompatibilities render most or all hybrids infertile or inviable[68]. In isolated insular biota, the odds that colonizing lineages will meet that have the right genetic distance, and hence the opportunity for hybridization to jump-start adaptive radiations, are slim and in many island lineages adaptive radiations never occurred. For these reasons, it remains an open question whether adaptive radiation is primarily limited by ecological constraints associated with niche preemption and competition when multiple related species colonize, or by genetic constraints associated with lack of genetic variation which can be overcome by hybridization when related species colonize the same new adaptive zone, but not when only one lineage colonizes.

To address this question, we studied the haplochromine cichlid assemblages in two geographically close and climatically similar East African great lakes. Both lakes are expected to provide ecological opportunity for adaptive radiation[12] but whereas Lake Mweru was colonized by Congolese and Zambezian species of several haplochromine cichlid lineages (and many other fish), Lake Bangweulu remained more biotically isolated having been colonized from the Zambezi only. In the melting pot Lake Mweru ecological opportunity (albeit possibly reduced by the larger number of colonizing fish species) hence coincided with opportunity for hybridization between previously allopatric lineages and four adaptive radiations evolved, all of admixed ancestry. In Lake Bangweulu where we saw no evidence of hybridization between the exclusively Zambezian lineages of haplochromine cichlids that occupy this lake, no adaptive radiation occurred despite ecological opportunity. Finally, in rivers of the Mweru drainage, where ecological opportunity for cichlid adaptive radiation is low[17], we found widespread signatures of hybridization but—with one exception—no radiation, suggesting that hybridization alone is not sufficient for adaptive radiation. The exception is the emergence of a new hybrid clade between two quite distantly related lineages (*Orthochromis*, *Pseudocrenilabrus*) that invaded extensive flooded still-water habitats in the Kalungwishi and Luongo river basins. All this is consistent with the hypothesis that rapid adaptive radiation requires the

coincidence in space and time of ecological opportunity and genetic admixture between lineages (additionally to other known factors that are present in all lineages we studied here, such as ecological versatility, sexual selection and a sexually dimorphic mating trait system[12]). The smaller area of Lake Bangweulu compared to Lake Mweru is expected to result in lower species richness in Lake Bangweulu, but cannot explain the complete absence of any in situ speciation because lake area does not predict whether or not cichlid fish adaptive radiations happen. Lake depth does predict adaptive radiation in cichlids in general, but the effect of lake depth as a predictor of adaptive radiation emerges strongly only at a scale of depth variation much larger than the difference in depth between our two lakes and we do observe radiations even in lakes much smaller and shallower than Lake Bangweulu[12]. The major difference between the two lakes therefore appears to lie in the number and geographical origins of colonizing lineages affecting both competition and opportunity for hybridization. We argue that in the absence of admixture variation kickstarting adaptive radiation[3,7], there was probably not enough time for accumulation of genetic variation through mutations in less than a million years in Lake Bangweulu.

Lake Mweru provides a rare opportunity to study interactions between sympatric and contemporaneously evolving radiations. We find evidence for both positive and negative effects of ecological lineage interactions on diversification. The two radiations of piscivores were possibly facilitated by the radiation of the genus *Pseudocrenilabrus* into diverse habitats including very small offshore forms. However, the astonishing eco-morphological complementarity among all Lake Mweru radiations is consistent with the hypothesis that they may also have constrained each other through competition and niche preemption. Morphological complementation can arise through competitive co-evolution and/or through historical contingency and genetic, developmental or constructional constraints[69,70]. Given the very different starting positions of the radiating lineages in morphospace, contingency is probably important. However, contingency did not prevent the radiation of serranochromines in the paleolake Makgadikgadi from generating morphological diversity that largely parallels the radiations of Lake Victoria and Malawi which started from very different ancestral conditions[42]. Yet, this did not happen in individual radiations in Lake Mweru. Conversely, contingency alone cannot readily explain the precise complementation between radiating lineages in Lake Mweru and their trajectories of morphospace expansion into space not occupied by the other radiations. Furthermore, large parts of the Makgadikgadi serranochromine morphospace that serranochromines in Lake Mweru lack are exactly those occupied in Lake Mweru by members of the "orthochromine" radiations. Secondly, *Pseudocrenilabrus* seem to have confined *Orthochromis* in Lake Mweru to the niche of rocky shore Aufwuchs feeders. That *Orthochromis* can evolve insectivorous soft bottom ecomorphs became surprisingly apparent to us when we found two *Pseudocrenilabrus*-like species in lake-like soft bottom habitats of the Kalungwishi and Luongo Rivers that are mitochondrially sister to the *O. kalungwishiensis* complex (Fig. 3c) and in the nuclear genome admixed with *Pseudocrenilabrus* (Supplementary Data 8, 9).

In conclusion, the colonization of Lake Mweru by several related but previously allopatric lineages of haplochromines with hybridization between them seems to have facilitated adaptive radiations in this lake. The absence of radiations in Lake Bangweulu may be explained by lack of hybridization (despite ecological opportunity). The absence of adaptive radiations in the rivers around Lake Mweru may be explained by the lack of ecological opportunity (despite hybridization). This is consistent with coincidence of ecological opportunity and genetic opportunity for hybridization being required for rapid adaptive radiation

(besides the other prerequisites that are given in haplochromines as explained above but not in many of the other fish lineages in these lakes). Our study suggests that despite diminishing ecological opportunity through competition, the presence of related lineages can facilitate adaptive radiation through increased genetic potential to respond to ecological opportunity. It thus shows that adaptive radiation does not require strict biotic isolation after first colonization but instead can be fuelled by simultaneous or successive colonizations by related lineages if these subsequently hybridize. Our study explicitly compares all adaptive radiations and all related lineages that did not radiate within the same system. More such studies are needed to understand how generalizable our results are.

## Methods

**Fieldwork.** Between September and October 2005, we sampled 11 locations along the southern and eastern coasts of Lake Mweru. Three of these sites were sampled again in 2017. Sampling sites included shallow, sandy beaches in the south, a rocky outcrop on Kilwa Island in the south-west, steeply sloping, wave exposed beaches and rocky boulder shores in the northeast, and offshore open waters in the south, south-west and northeast. We also sampled two sites in the large lagoon network directly south of Lake Mweru. In 2005, we sampled the lower reaches of the Kalungwishi River. In 2017, we sampled five sites in the middle and upper reaches of the Kalungwishi River. In Lake Bangweulu, we sampled five locations, including white sandy beaches, large water lily beds and extensive reed swamps. Catching methods included monofilament gill nets and beach seines operated by teams of professional fishermen. Other fish were obtained from angling youth at rocky shores, from offshore commercial gillnet fisheries, and from one large commercial landing site in each lake. We collected and preserved a total of 404 specimens from Lake Bangweulu, more than 1000 specimens from Lake Mweru and about 50 specimens from the rivers. These were first preserved in formalin and then transferred to 75% ethanol in the laboratory. Fish are held in the collection of the University of Lusaka, Zambia, and at EAWAG, Switzerland.

**Mitochondrial DNA analysis.** We sequenced the mitochondrial control region (D-loop) of 223 individuals and added 199 sequences from GenBank (a detailed list of all specimens including sampling localities and Genbank accession numbers is provided in Supplementary Data 11). Fin clips were preserved in 95% ethanol. DNA was extracted using proteinase K digestion and Promega Wizard DNA extraction kit (Promega) following the manufacturer's protocol. All samples were standardized to a DNA concentration of 50 ng/µl.

The Mitochondrial control region was amplified with polymerase chain reaction (PCR) using the forward primer Fish L15926, 5′-GAG CGC CGG TCT TGT AA-3′ and the reverse primer Fish 12s, 5′-TGC GGA GAC TTG CAT GTG TAA G-3′. The PCR program included initial denaturation at 94 °C for 3 min followed by 35 cycles of denaturation at 94 °C for 45 s, annealing at 56 °C for 45 s, extension at 72 °C for 1.30 min and final extension at 72 °C for 5 min. PCR products were cleaned using exonuclease and shrimp alkaline phosphatase (ExoSAP). DNA fragments were amplified for sanger sequencing using BigDye Terminator (ABI) according to the manufacturer's instructions, adding 1 M betaine to the sequencing reaction. Sanger sequencing was performed using an ABI capillary sequencer[42]. Alignment of sequences was done in ClustalX, using pairwise alignment[71] and adjusted by eye.

**Phenotypic analysis.** To characterize the ecological and phenotypic diversity of Lake Mweru, we phenotyped 206 individuals of Lake Mweru and the Kalungwishi River including 36 *Sargochromis*, 34 large-tooth serranochromines, 34 small-tooth serranochromines, 73 *Pseudocrenilabrus* and 23 *Orthochromis* samples (Supplementary Data 12). We measured 13 standard morphometric distances with digital callipers to the nearest 0.01 mm, the combination of which is powerful to detect even fine eco-morphological differences between species[41,72]. We used between one and 15 individuals per putative species from Lake Mweru. The traits[41] measured were: standard length (SL), body depth (BD), head length (HL), head width (HW), snout length (SnL), snout width (SnW), lower jaw length (LjL), lower jaw width (LjW), eye length (EyL), eye depth (EyD), interorbital width (IoW), cheek depth (ChD) and preorbital depth (PoD). Trait variables were log-transformed to homogenize variance and subsequently size-corrected by performing pooled within-group regressions (species) of the characters against standard length. The standardized residuals of the linear regression were used in a principal component analysis (PCA) to retain the shape differences between the candidate species.

A PCA was run with the R-function 'prcomp' for all radiations together to assess the morphospace used, and for each radiation separately, to assess differences between candidate species. To ask if candidate species were phenotypically differentiated, one-way ANOVAs were calculated on the first three principle components with 'candidate species' as factor.

We also compared the total morphological diversity of the Lake Mweru haplochromine radiations to representatives of the putative ancestral lineages in

Lake Mweru (six samples) and haplochromine cichlids in Lake Bangweulu (34 samples) to assess the morphospace expansion during the radiation. Here, the size correction was performed by genus instead of species. In addition, we compared Lake Mweru haplochromines to previously published[42] representatives of the serranochromine radiation from the paleo-Lake Makgadikgadi (94 samples) and four *P. philander* of Lake Bangweulu. Because measurements of EyD, POD, SnW and HW were not available for a large proportion of Makgadikgadi and some Bangweulu samples, the PCA including these samples were performed with the remaining eight traits. To compare the potential morphospace occupation of the serranochromines with the real morphospace occupation in Lake Mweru, we predicted all Lake Mweru samples into the morphospace computed with paleo-Lake Makgadikgadi samples using the 'predict' R function.

**RADseq data generation.** Restriction-site associated DNA sequencing (RADseq) was performed with a strategically selected subset of Lake Mweru and Bangweulu samples and multiple outgroups to check if phenotypically defined candidate species were monophyletic and how they phylogenetically relate to each other (175 samples, Supplementary Data 13). We mostly followed the standard protocol by Baird et al.[73] with minor modifications: Restriction digestion was performed overnight using the restriction enzyme *Sbf*I (NewEngland Biolabs). Each RAD library contained 48 individuals each tagged with a unique barcode of 5–8 bp length. Each RAD library was sequenced single-end on an Illumina HiSeq 2500 lane at the NGS Center of the University of Bern or at the Lausanne Genomic Technologies Facilities. Some individuals were sequenced twice to increase sequencing depth.

The reads were trimmed to 84 bp and demultiplexed using the process_radtags script from Stacks[74] allowing only a single error in the restriction site and barcode. Reads containing at least one base with a Phred quality score below 10 or more than 10% of bases with quality below 30 were filtered with the FastX toolkit (http://hannonlab.cshl.edu/fastx_toolkit). The reads were then end-to-end aligned to the *Oreochromis niloticus* reference genome[75] using Bowtie2 v. 2.2.6[76]. We called variants and genotypes with Unified Genotyper v. 3.7 (ref. [77]). We removed sites with more than 50% missing data, a mean sequencing depth greater than 47 reads, or heterozygosity exceeding 50%. For analyses requiring only variants, we extracted bi-allelic SNPs with a minimum quality of 30 and removed genotypes with a minimum genotype quality of 15.

**Genomic structure of the radiations.** PCAs were performed with the R package 'SNPRelate' on SNP datasets for each radiation separately, and for habitat groups of the *Pseudocrenilabrus* radiation to assess differentiation between candidate species. Only sites with maximum 20% missing data and a minor allele frequency of 5% were used. We performed analyses of variance (ANOVA) with the 'anova' R function using species as factors and PC axes as dependent variables to assess if our candidate species explain the observed genetic variation.

We performed an ADMIXTURE analysis[55] to infer groupings and cluster assignments of serranochromines. We performed fivefold cross-validation error estimation to determine the best number ($k$) of clusters.

**Phylogenomic analyses and dating of phylogenetic trees.** We used RAxML v. 8.2.9 to generate phylogenetic trees using the GTRGAMMA model and 100 bootstrap replicates for the RAD data and the mitochondrial data separately. The tree was rooted with five Lamprologini samples (*Neolamprologus* and *Lamprologus* spp.). To compare the timing of the onset of the four radiations in Lake Mweru, we ran BEAST2 analyses[78] using the mitochondrial dataset. We estimated the best model with jModelTest v 2.1.10[79] and used the estimated mutation rates as priors for the BEAST2 analysis. To reduce the dataset and optimize symmetry of the tree, we selected six haplotypes of each radiation represting the entire haplotype diversity, and a subset of the closest relatives of the radiations. We additionally added samples from Lake Malawi and the Lake Victoria Region for calibration and used four different calibration sets as detailed in Supplementary Table 1. We ran BEAST2 with a Calibrated Yule Model using a relaxed log normal clock. The MCMC chain length was set to 10 million with a pre-burnin of 100,000 and trees were logged every 20,000 steps. We used TreeAnnotator of the BEAST2 package to compute confidence intervals for the radiation nodes. We checked if all ESS values were above 200 and if runs had converged with Tracer v1.6[80]. We visualized the calibrated tree with FigTree v1.4.3 and subsequently simplified it in Inkscape v. 0.9.2.

**Hybridization analyses.** To identify hybridization between different groups in Lake Mweru or Bangweulu, we ran a series of D statistics and f4 tests with the 'qpDstat' tool of ADMIXTOOLS v. 701[53]. In order to assess the robustness of D statistic results, we ran each test with several different outgroups. The different combinations of taxa tested are shown in Supplementary Data 2–5 for serranochromines and in Supplementary Data 7–8 for "orthochromines". Tests focusing on Lake Bangweulu taxa are shown in Supplementary Data 10. For "orthochromines", f4 tests revealed less than 50 ABBA or BABA patterns in almost all tests and are thus not shown even though they support the hybridization events detected with D statistics. In addition to D statistics, we ran MixMapper v. 2.0[54] on specific groups with evidence for admixture. MixMapper first infers a backbone tree of reference groups provided by the user and in a second step infers the ancestry of a

putatively admixed test group as a mix of two of the reference groups by solving a series of equations based on f2 and f3 tests. We performed 100 bootstrap replicates with blocks of 50 SNPs. All taxa tested of serranochromines are shown in Supplementary Data 6, and of "orthochromines" in Supplementary Data 9. If not indicated otherwise, all taxa were provided as potential ancestor proxies. Mix-Mapper v. 2.0 also allows for three-way admixture models, whereby a taxon is modelled as admixed between an ancestor that is itself admixed and an additional ancestor proxy. For some taxa where such a scenario seems likely, we tested if a three-way admixture model fits the data better than a simple two-way admixture model using the "options.test_2v3" implemented in MixMapper v .2.0.

To test wether there is evidence for recent gene flow, we used fineRADstructure[81]. We phased the data with the GATK tool ReadBackedPhasing[77]. Next, we defined the RADtags using the read position information in the bam files and discarded RADtags found in less than ten individuals or with less than 50 sites. This resulted in a dataset of 23,060 RAD loci. Finally, we converted the phased vcf file to the "Chromosome + Tag Haplotype Matrix" format required by fineRADstructure using a custom script (vcf2fineRADstructure.sh provided on https://github.com/joanam/scripts). We ran fineRADstructure on the datasets of "orthochromines" and serranochromines separately. We excluded individuals with more than 50% missing data and downsampled the *Pseudocrenilabrus* radiation to 20 individuals with least missing data.

**Ethical declaration.** This project was carried out in compliance with all relevant ethical regulations for animal research.

**Reporting summary.** Further information on research design is available in the Nature Research Reporting Summary linked to this article.

## Data availability
Mitochondrial sequences newly generated for this study have been uploaded to GenBank with accession numbers MN167933–MN168155. Raw reads of RAD data are available at the NCBI Sequence Read Archive under Bioproject PRJNA553794. The morphology data, the D statistics used for plotting, the genomic data files used for generating PCA plots, the fineRADstructure output files, the BEAST trees with all calibration sets, the RAD variant calls, and the mitochondrial and RAD RAxML trees are provided at Zenodo at https://doi.org/10.5281/zenodo.3435419. The fully labelled trees underlying Supplementary Fig. 2 are given as Supplementary Data 1. All D statistics, f4 test results and MixMapper results are given as Supplementary Data files 2–10. Information on the samples used for mitochondrial sequencing, morphological measurements and RAD sequencing are provided as Supplementary Data 11, 12 and 13, respectively.

## Code availability
The R scripts underlying Figs. 2–4, Supplementary Figs. 4, 5, 8, 9, and the phylogenies of Supplementary Data 1 are available at Zenodo at https://doi.org/10.5281/zenodo.3435419. File conversion scripts are provided on GitHub (https://github.com/joanam/scripts) as indicated in the main text.

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

## Acknowledgements

We thank C. Kapasa and P. Ngalande of the Department of Fisheries in the Ministry of Fisheries and Livestock of Zambia for their support during fieldwork, and Lackson Shiridzinodya, Joseph Chishimba and Happy Tembo for their help in the field. We thank village headman Paul Chilufya for his eye-opener present of colourful offshore *Pseudocrenilabrus* from his net. Thanks to J.P. Danko for critical review of early versions of this manuscript, to Curt Stager for a stimulating discussion, and to Hans van Heusden for providing samples and photographs of river cichlids. Genomic analyses were performed using the computing infrastructure of the Genetic Diversity Center (GDC) at ETH Zürich. This research was supported by Swiss National Science Foundation grants 3100A0-106573, ProDoc PDFMP3_134657 and 1535 17-870, all to O.S.

## Author contributions

J.I.M. prepared RAD libraries, analysed RAD and mitochondrial sequence data, performed phylogenetic and hybridization analyses, analysed morphological data, made all figures, and wrote the manuscript with O.S. and R.B.S.; R.B.S. did fieldwork, collected mitochondrial sequence and AFLP data, collected morphological data, analysed data, wrote the manuscript with O.S. and J.I.M.; C.K. coordinated and implemented fieldwork, collected morphological data, collected mitochondrial sequence data; S.M. prepared RAD libraries, collected mitochondrial sequence data and supervised lab work; O.M.S. analysed morphological data; D.A.J. did fieldwork, supervised lab work; N.P. did fieldwork and DNA extractions; C.E.W. analysed data; U.K.S. provided and identified samples from rivers and contributed to taxon sampling discussion; O.S. wrote the grant, designed the project, coordinated and implemented fieldwork, identified all fish, collected morphological data, analysed data, supervised the project, and wrote the manuscript with R.B.S. and J.I.M.; all authors commented on drafts and approved of the final version.

## Competing interests

The authors declare no competing interests.
