## [Peer Review File · Nature Communications]

Reviewers' Comments:

Reviewer #1:

Remarks to the Author:

This is a very nice and important piece of work documenting the role of hybridization in cichlid fish radiations. The authors sought to tease apart the question of whether adaptive radiation is limited by ecological (niche preemption, competition) constraints or alternatively by genetic constraints (limited genetic variation). They examine this by comparing two lakes, Lake Mweru and Lake Bangweulu, which are geographically close and climatically similar. The primary difference is that Lake Mweru was colonized by from multiple sources, while Lake Bangweulu was colonized by a single source. The particular significance is that the authors are able to tease apart the precise history of hybridization in Lake Mweru, and hence its role in feeding the genetic diversity in a given lake, while also comparing this to a lake system within which hybridization has not occurred.

The authors do a great job of detailing the hybridization events in Lake Mweru. Somewhat less convincing was the comparison with Lake Bangweulu in the context of ecological opportunity. Ecological opportunity refers to the resources that can be exploited as a result of a combination of isolation (reduces the number of competitors/ predators, etc) and resources available. The opportunity for adaptive radiation appears to require both these factors; thus, in terrestrial landscapes, adaptive radiation has not been documented on the most isolated atolls, presumably due to the limited area and resource availability. For the current study, the comparison is between Lake Bangweulu and Lake Mweru, with the idea that Lake Bangweulu should provide greater opportunity because it is more isolated than Lake Mweru. The problem here is twofold: (1) the authors do not talk about what factors were used to define "isolation", and hence how Bangweulu is considered to be more isolated than Mweru. Moreover, (2) there is no information provided on the relative volumes of these lakes, which seems to be critical to the manuscript given the test of the role of ecological opportunity. In the supplemental material the authors say that Lake Bangweulu has an area of 3,000 km² surface (expanding in the wet season), max depth 10 m; Lake Mweru has an area of 4,400 km² surface, max depth 27 m. I could not find references to these values given in the text. However, in checking Shiklomanov and Rodda (2004), these authors give Bangweulu as having an area of 4920 km² surface area, depth 5 m and volume 5 km³; Lake Mweru having an area of 5,100 km² surface, depth 15 m, and volume 32 km³. If this is correct, then - although the areas are roughly comparable - the huge differences in volumes would mean that Lake Mweru would likely provide much more opportunity compared to Lake Bangweulu. Therefore, could not the finding that diversity is much higher in Lake Mweru be explained simply by the much greater lake size (volume)?

The real strength of the paper is the novelty of the adaptive radiation in Lake Mweru and the history of hybridization that has fed into this lake. I think that the paper would be a significant contribution if the authors focused on this element. Yes, it would be great to tease apart the effects of ecological and genetic opportunity, but I'm not convinced that this system is the best for doing that.

Specific points:

Line 80: To what does "They" refer?

"... provide ecological opportunities such as opportunities .." Rerword

Line 82 "Here, we test if co-occurrence of multiple lineages facilitated or impeded .." Not clear from this as to the goal. If we're thinking 2-3 lineages, the answer would be very different if we're thinking 500k - 10,000k ..

Line 84 Why cichlids?

Line 99 "despite the presence"

Line 100 "favorable conditions for radiation (e.g. lake size, latitude, .." Clarify. What about lake size,

latitude, etc, makes it favorable (presumably not small size and high latitude)

Line 114. I am not sure that the large differences in lake volume (and presumed associated ecological opportunity) allow you to separate out this particular effect. As mentioned above, there is no definition of isolation, so it is difficult to see how this differs between the lakes. Moreover, the volume of Lake Mweru seems to be much larger than Bangweulu, so the prediction based on this alone would be that the Mweru would have a much greater diversity of species (as reported).

Line 136 Not clear as to the significance of the point that, while > 600 individuals were caught, only 400 were collected; likewise > 1200 compared to 1066

Line 192. Not clear what is complementary to what.

Line 199 (& 255). What are "Aufwuchs"

Line 241: Overlap in species?

Lines 251-269 - Patterns of diversity in *Orthochromis* and *Pseudocrenilabrus* are suggestive of evolutionary effects of competition in sympatry because in sympatry the two lineages do not overlap in morphospace, but the "*Orthochromis*" in allopatry overlaps in morphospace tremendously with *Pseudocrenilabrus*. That's fine. Certainly consistent with competition, but there is no actual evidence of competition. So the statement that competition among lineages appears to have constrained the adaptive radiations in Lake Mweru seems highly speculative. Likewise that interactions among the most distantly related lineages may have facilitated radiation.

Line 310 serranochromines

Line 411: occurred

Line 413: Keep in mind the work on many plant radiations that indicate both early and ongoing hybridization in adaptive radiation.

Line 441. "despite ecological opportunity" – Looks like there would be a lot less ecological opportunity in Bangweulu given the much smaller volume?

Line 456 Say what is complementary to what.

Line 482. The evidence is really not sufficient to say that the opportunity for hybridization is more important than ecological opportunity.

Reviewer #2:

Remarks to the Author:

In this interesting paper, Joana Meier and co-workers provide an in-depth examination of the cichlid fish faunas of two so far underexplored African Lakes, Lake Mweru and Lake Bangweulu, which are situated at a biogeographically interesting region between the Congo and the Zambezi drainage systems. Based on an impressive sample size, the authors report a major difference between these two lakes with respect to cichlid adaptive radiation: While there are only few and usually non-endemic cichlid species in Bangweulu, Lake Mweru is home to a large species assemblage mainly derived from intralacustrine radiation events. The authors relate this pattern to the much higher potential for (historical) gene flow between species in Lake Mweru. This observation is not particularly novel, given that the same authors have previously shown that hybridization can fuel adaptive radiation. What is new in this study is the suggestion that the co-existence of several closely related lineages in the same water body is conducive for radiation (because of the possibility to exchange genes) rather than

preventing it (as would be expected from ecological theory).

Overall, I find this integrative study very well conducted, the methods are sound, and the results are of broad interest. I do have a few points, though, which would need further clarifications or which the authors should consider to incorporate in a revised version of their work:

(1) I do think that the authors should not dismiss the differences between the two lakes in their consideration of whether or not an adaptive radiation takes place (that is, the abiotic factors). Lake Mweru is certainly larger, probably more stable over geological times, it is deeper (so there is real deep-water/open water niche), and it is situated at a lower altitude (these are all factors that contribute to adaptive radiation according to previous work by the same group; see Wagner et al.). Bangweulu, on the other hand, is a swamp after all. While I don't think that these abiotic factors are crucial for the story, these should be discussed in more detail.

(2) Although not particularly important for the general interpretation of the results, I did not get the point why exactly the four age calibrations presented in Table S1 were chosen, as these do not include the minimum possible, nor the most up-to-date or the ones based on the largest data-sets. I would suggest to have a look at Michael Matschiner's review on that matter (*Hydrobiologia* cichlid special issue). Wouldn't it be possible to also try a demographic estimate of diversification times?

(3) As mentioned above, I am generally quite happy with the methods. Nevertheless, I thought that the data-set at hand would ideally be run through Milan Malinsky et al.'s *fineRADstructure* software, which is specifically tailored for the RAD data such as the ones presented in this study. Such an analysis would allow for a detailed view on the co-ancestry of the taxa.

(4) I feel that the authors should go over the taxonomy once more and provide updates, where necessary. For example, I believe that the taxon labelled as "New Lufubu" is in fact *O. indermauri* (Schedel et al. 2018); and perhaps some other taxa have been described (I might be wrong, though, but was not able to trace this back further).

In addition, I have some rather minor comments/suggestions:

- 95: I don't want to advertise work on which I am co-author, but I think that Britta Meyer's PhD work (the *Systematic Biology* paper) is relevant here.
- 108/109: Out of interest: How do we know that *Tylochromis* is a strong swimmer compared to other cichlids? Is there a reference for this statement?
- 239: there's a double space before "paleolake"
- 279: To be fair, I think you should cite the original papers from where this number comes from, the Nagel paper (200k) and the Verheyen paper, which first reported this exact number (again, I am a co-author on the latter)
- 310: Chilwa or Chila? I think this makes a big difference (see also our latest work on sex chromosome evolution in *P. philander*...)
- 476, etc.: very good point.
- 646: AFLP's: I see the point that you don't have the extracts anymore, but is it really necessary to report these outdated results and methods?
- It would be great if you could improve a bit the .pdf of the detailed nuclear versus mitochondrial tree. Node labels, symbols and especially the branch support values are really difficult to read.

I very much hope that my comments are useful.

With best wishes,

Walter

Reviewer #3:

Remarks to the Author:

I read carefully the manuscript entitled "The coincidence of ecological opportunity with hybridization explains the prevalence of rapid adaptive radiation". This study showed that adaptive radiations of four haplochromine lineages from Lake Mweru experienced hybridization in ancestral species with other cichlid lineages, but non-radiated haplochromine lineages from Lake Bangweulu haven't experienced hybridization. Ecological opportunity for adaptive radiation may not be different so much between the lakes; therefore, not only ecological opportunity but also hybridization is likely to be a key factor facilitating adaptive radiation. I didn't find problems in analyses and conclusion but very minor points regarding explanation.

Line 151-154: This study called Pseudocrenilabrus radiation, Sargochromis radiation, large-tooth serranochromines, and small-tooth serranochromines "lacustrine radiations" of Lake Mweru. This information is very important in this study, but I didn't understand which are lacustrine radiations when I read the text first. Showing these radiations in this sentence, or marking these radiations with "L" or some character in Figure 1b will be helpful for readers.

Line 192: It is difficult to follow this section. For example, I confused "Orthochromis species (line 198)" with orthochromines species at first. In the previous section, eight radiations were defined. In this section, explanation for each radiation may be easier to understand.

Point-by-point response to reviewers' comments:

Reviewer #1:

This is a very nice and important piece of work documenting the role of hybridization in cichlid fish radiations. The authors sought to tease apart the question of whether adaptive radiation is limited by ecological (niche preemption, competition) constraints or alternatively by genetic constraints (limited genetic variation). They examine this by comparing two lakes, Lake Mweru and Lake Bangweulu, which are geographically close and climatically similar. The primary difference is that Lake Mweru was colonized by from multiple sources, while Lake Bangweulu was colonized by a single source. The particular significance is that the authors are able to tease apart the precise history of hybridization in Lake Mweru, and hence its role in feeding the genetic diversity in a given lake, while also comparing this to a lake system within which hybridization has not occurred.

The authors do a great job of detailing the hybridization events in Lake Mweru. Somewhat less convincing was the comparison with Lake Bangweulu in the context of ecological opportunity. Ecological opportunity refers to the resources that can be exploited as a result of a combination of isolation (reduces the number of competitors/ predators, etc) and resources available. The opportunity for adaptive radiation appears to require both these factors; thus, in terrestrial landscapes, adaptive radiation has not been documented on the most isolated atolls, presumably due to the limited area and resource availability. For the current study, the comparison is between Lake Bangweulu and Lake Mweru, with the idea that Lake Bangweulu should provide greater opportunity because it is more isolated than Lake Mweru.

>> We are glad that the reviewer finds our manuscript interesting and important. We would like to thank the reviewer for the constructive and helpful comments. We address the author's point about the manuscript needing expansion in the comparison of ecological opportunity in Lake Mweru and Lake Bangweulu below.

1.1. The problem here is twofold: (1) the authors do not talk about what factors were used to define "isolation", and hence how Bangweulu is considered to be more isolated than Mweru.

>> Lake Bangweulu is more isolated biotically because it used to be part of the uppermost Zambezi drainage system and even though it is now part of the Congo drainage system, waterfalls have prevented colonization by Congolese haplochromines and most other Congolese fish. Thus the only source of lineages in this lake is the old upper Zambezi system, and there is many fewer fish species occurring in this lake than in Lake Mweru. In contrast, Lake Mweru is connected both to the Congo and the Zambezi drainage systems. It used to be part of the Congo and originally hosted a Congolese fish fauna. More recently it received massive input of Zambezian fauna from Lake Bangweulu through the Luapula river capture event. Downstream movement of Zambezian lineages from Lake Bangweulu into Lake Mweru is possible, but upstream movement of Congolese lineages from Lake Mweru into Lake Bangweulu is impeded by waterfalls and extensive rapids in the Luapula River.

To make these connections clearer we replaced "isolation" by "Greater biotic isolation with fewer lineages present" in the abstract (L. 57) and write "biotic isolation" instead of just "isolation" in the main text. In the Introduction we now write: "Waterfalls and extensive rapids make upstream fish movement

of haplochromines and many other fish from Lake Mweru to Lake Bangweulu very difficult, but do not fully preclude downstream colonization of Lake Mweru by Zambebian lineages from Lake Bangweulu. The cichlids of Lake Bangweulu have thus been more isolated than those in Lake Mweru in terms of connections of drainage systems.” (L. 116).

1.2 Moreover, (2) there is no information provided on the relative volumes of these lakes, which seems to be critical to the manuscript given the test of the role of ecological opportunity. In the supplemental material the authors say that Lake Bangweulu has an area of 3,000 km² surface (expanding in the wet season), max depth 10 m; Lake Mweru has an area of 4,400 km² surface, max depth 27 m. I could not find references to these values given in the text. However, in checking Shiklomanov and Rodda (2004), these authors give Bangweulu as having an area of 4920 km² surface area, depth 5 m and volume 5 km³; Lake Mweru having an area of 5,100 km² surface, depth 15 m, and volume 32 km³. If this is correct, then - although the areas are roughly comparable - the huge differences in volumes would mean that Lake Mweru would likely provide much more opportunity compared to Lake Bangweulu. Therefore, could not the finding that diversity is much higher in Lake Mweru be explained simply by the much greater lake size (volume)?

>> The reviewer raises an important point for which we have now added more information to the main text so that readers do not have to refer to the Supplementary Notes anymore for this important piece of information. We have also added references to the numbers reported in the Supplementary Notes. Lake Mweru’s greater depth is confined to a very small area and the rest of the lake is similar in depth to Lake Bangweulu (Bos, et al. 2006). The difference in volume is thus smaller than the reviewer assumes. We do agree that Lake Mweru is larger than Lake Bangweulu. However, lake size is not a predictor of whether cichlid adaptive radiations evolve in African Lakes (see Wagner et al 2012). Lake area does predict species richness, but not at all whether or not radiations occur (see Wagner et al. 2012). Maximum lake depth predicts whether radiation happens, but the depth of Bangweulu lies well within the depth of lakes that do support endemic radiations elsewhere in Africa (Wagner et al. 2012).

We now write on L. 100 - 112: “Lake Mweru is larger than Lake Bangweulu. It has a surface area of 5,100 km² (Shiklomanov and Rodda 2004) with a mean depth of 7.5 m and a maximum depth of 27 m (Bos, et al. 2006). The surface area of Lake Bangweulu is subject to strong seasonal changes and is reported as 2,500-5,000 km² (Herdendorf 1982; Bos and Ticheler 1996; Shiklomanov and Rodda 2004), with a mean depth of 4.7 m and a maximum depth of 10.4 m (Hughes 1992). Previous work shows that for African lakes in general, lake depth predicts adaptive radiation of cichlids, along with sexual dichromatism and energy measured as solar radiation (Wagner et al., 2012). Lake surface area, on the other hand, does NOT predict whether a radiation occurs. Lake surface area predicts the species richness of cichlid assemblages both when radiations happened and when not, whereby lakes with radiation have much greater species richness per area than lakes without radiation (Wagner et al., 2014). In fact, radiations are known from lakes much smaller than Lake Bangweulu (Wagner et al., 2014). Concluding from these previous analyses, the greater depth likely facilitates radiation in Lake Mweru, but the parameters of Lake Bangweulu are similar and lie within the range that permitted in situ adaptive radiation in lakes elsewhere.”

1.3 The real strength of the paper is the novelty of the adaptive radiation in Lake Mweru and the history

of hybridization that has fed into this lake. I think that the paper would be a significant contribution if the authors focused on this element. Yes, it would be great to tease apart the effects of ecological and genetic opportunity, but I'm not convinced that this system is the best for doing that.

>> We agree that the discovery of multiple recent radiations in Lake Mweru and the important role of hybridization in this lake is the most important finding of our study. We think that the comparison of these radiations in Lake Mweru to the lack of adaptive radiations in the same lineages in Lake Bangweulu where no hybridization occurred and the lack of adaptive radiations in again the same lineages in rivers where no ecological opportunity exists strengthens our finding that the coincidence of ecological opportunity and hybridization is most conducive to adaptive radiation.

Differences in lake size may affect how many species evolve in a cichlid radiation but earlier work has shown that they do not explain whether or not radiation happens, and the complete absence of radiations in Lake Bangweulu cannot be explained by the lakes physical attributes. We now discuss this in more detail: "The smaller size of Lake Bangweulu compared to Lake Mweru is expected to result in lower species richness in Lake Bangweulu, but cannot explain the absence of radiations. Lake area does not predict whether or not cichlid fish adaptive radiations happen, and many lakes much smaller than Lake Bangweulu host adaptive radiations (Wagner, et al. 2012). We argue that in the absence of admixture variation kickstarting adaptive radiation (Seehausen 2004; Marques, et al. 2019), there was probably not enough time for radiation through accumulation of mutations in less than a million years." (L. 483-489). We have also rewritten the conclusions paragraph to focus more on the hybrid origin of the radiations in Lake Mweru and less on the direct comparison to Lake Bangweulu in response to the author's suggestion that this subject be more clearly the main focus of the paper.

Specific points:

1.4 Line 80: To what does "They" refer?

" ... provide ecological opportunities such as opportunities .." Reword

>> We replaced "they" by "divergent lineages" (L. 80)

1.5 Line 82 "Here, we test if co-occurrence of multiple lineages facilitated or impeded .." Not clear from this as to the goal. If we're thinking 2-3 lineages, the answer would be very different if we're thinking 500k – 10,000k.

>> We are not sure what the reviewers means here. The comment refers to the following sentence: "Here, we test if co-occurrence of multiple lineages facilitated or impeded adaptive radiation in African cichlid fish." Here, we lay out the general goals and do not want to already give exact numbers of lineages. Lake Mweru has overall many more immigrant fish species, twice more haplochromine cichlids, twice more other cichlids, and many more (probably at least twice more) other fish.

1.6 Line 84 Why cichlids?

>> On that line we wrote "Cichlids have radiated in more than 30 lakes in Africa (Wagner et al., 2012)." Does the reviewer mean why cichlids are the ones that radiate in all of these lakes and not the other fishes? Or is he/she suggesting that we focus more broadly on discussing adaptive radiations? In either

case, we think that these subjects go beyond the scope of our manuscript. In terms of why cichlids have radiated, as Wagner et al. (2012) showed, lake and lineage-specific factors influence the likelihood of adaptive radiation: cichlids radiate more often when they experience sexual selection and when they occur in deeper lakes with higher solar energy and fewer predators. However, there is considerably variation left unexplained by the macroevolutionary model of Wagner et al. (2012), such as the large difference between the nearby lakes Mweru (4 radiations) and Bangweulu (no radiation), whose lineages share all the same biological traits and whose lake environments are similar. For the present paper we therefore chose to compare cichlid evolution in these two lakes. We provide evidence for an additional important factor that may be of general importance in rapid species radiations: large genetic variation generated through hybridization.

1.7 Line 99 "despite the presence"

>> Thanks for spotting this mistake. We have added "the"

1.8 Line 100 "favorable conditions for radiation (e.g. lake size, latitude, .." Clarify. What about lake size, latitude, etc, makes it favorable (presumably not small size and high latitude)

>> We now write "large lake size, low latitude..." (L. 99)

1.9 Line 114. I am not sure that the large differences in lake volume (and presumed associated ecological opportunity) allow you to separate out this particular effect. As mentioned above, there is no definition of isolation, so it is difficult to see how this differs between the lakes. Moreover, the volume of Lake Mweru seems to be much larger than Bangweulu, so the prediction based on this alone would be that the Mweru would have a much greater diversity of species (as reported).

>> We now provide more information on this topic in the Introduction and Discussion to better explain and discuss the differences between the two lakes (see our replies to comments 1.2 and 1.3). We argue that the complete absence of radiations in Lake Bangweulu cannot be explained by the physical attributes of the lake. Much smaller and shallower lakes elsewhere in Africa have cichlid radiations.

1.10 Line 136 Not clear as to the significance of the point that, while > 600 individuals were caught, only 400 were collected; likewise > 1200 compared to 1066

>> The first number refers to the number of fish caught and identified, the second number refers to what has been retained and preserved. We have now reformulated this sentence to make it clearer what these numbers refer to. E.g. "We caught and identified over 600 cichlids in Lake Bangweulu at 8 sites, of which we preserved 404." (L. 153)

1.11 Line 192. Not clear what is complementary to what.

>> We added "with each other" (L. 213)

1.12 Line 199 (& 255). What are "Aufwuchs"

>> “Aufwuchs” is a term commonly used in aquatic ecology and refers to small animals and plants that adhere to open surfaces in aquatic environments. We have now added an explanation in parenthesis: “Aufwuchs (periphyton and benthic invertebrates).” (L. 219)

1.13 Line 241: Overlap in species?

>> We have added “overlap in morphospace” to improve clarity.

1.14 Lines 251-269 - Patterns of diversity in *Orthochromis* and *Pseudocrenilabrus* are suggestive of evolutionary effects of competition in sympatry because in sympatry the two lineages do not overlap in morphospace, but the "*Orthochromis*" in allopatry overlaps in morphospace tremendously with *Pseudocrenilabrus*. That's fine. Certainly consistent with competition, but there is no actual evidence of competition. So the statement that competition among lineages appears to have constrained the adaptive radiations in Lake Mweru seems highly speculative. Likewise that interactions among the most distantly related lineages may have facilitated radiation.

>> We have tempered both statements by writing that the patterns are “consistent” with competition. Regarding the interactions among radiations of *Serranochromis* and *Pseudocrenilabrus*: We have added an additional sentence making it clearer that this is a speculation: “However, this hypothesis should be tested further with ecological analyses in the future.” (L. 293)

1.15 Line 310 *serranochromines*

>> We replaced “small-tooth radiation” by “small-tooth *serranochromines*” as the reviewer suggested.

1.16 Line 411: occurred

>> Typo corrected, thanks for spotting it!

1.17 Line 413: Keep in mind the work on many plant radiations that indicate both early and ongoing hybridization in adaptive radiation.

>> Yes, the citations here were indeed quite animal-biased. We have added seminal work on Hawaiian silverswords, *Howea* palms, Hawaiian *Cyrtandra*, sunflowers and Hawaiian mints. (L. 446)

1.18 Line 441. "despite ecological opportunity" – Looks like there would be a lot less ecological opportunity in *Bangweulu* given the much smaller volume?

>> We now discuss this important point at the end of this paragraph (L. 483-489). See our replies to comments 1.3.

1.19 Line 456 Say what is complementary to what.

>> In this paragraph, we discuss the “evidence for both positive and negative effects of ecological lineage interactions on diversification.” We had written “eco-morphological complementarity among Lake Mweru radiations”. Maybe it was unclear how this statement links to the previous sentence where

we discuss the predator-prey interactions between the piscivore radiations and *Pseudocrenilabrus*. To make it clearer that the “eco-morphological complementarity” is not just among these piscivores and *Pseudocrenilabrus*, we now write “eco-morphological complementarity among all Lake Mweru radiations” (L. 496).

1.20 Line 482. The evidence is really not sufficient to say that the opportunity for hybridization is more important than ecological opportunity.

>> We do not want to claim this and thank the reviewer for pointing out that this sentence could be misunderstood. In this sentence we compared the importance of the effects due to the presence of multiple closely related lineages: reduced ecological opportunity through competition versus increased genetic opportunity through hybridization. We are not comparing the relative importance of hybridization and ecological opportunity per se. Ecological opportunity is certainly required for adaptive radiation and thus of critical importance. We have reformulated this sentence to make this point clearer: “Our study suggests that despite potential reduction in ecological opportunity through competition among multiple lineages, the presence of related lineages can facilitate adaptive radiation through increased genetic opportunity. It thus shows that adaptive radiation does not require strict biotic isolation after first colonization but instead can be fueled by simultaneous or successive colonization of multiple related lineages if these subsequently hybridize.” (L. 523-529)

Reviewer #2:

In this interesting paper, Joana Meier and co-workers provide an in-depth examination of the cichlid fish faunas of two so far underexplored African Lakes, Lake Mweru and Lake Bangweulu, which are situated at a biogeographically interesting region between the Congo and the Zambezi drainage systems. Based on an impressive sample size, the authors report a major difference between these two lakes with respect to cichlid adaptive radiation: While there are only few and usually non-endemic cichlid species in Bangweulu, Lake Mweru is home to a large species assemblage mainly derived from intralacustrine radiation events. The authors relate this pattern to the much higher potential for (historical) gene flow between species in Lake Mweru. This observation is not particularly novel, given that the same authors have previously shown that hybridization can fuel adaptive radiation. What is new in this study is the suggestion that the co-existence of several closely related lineages in the same water body is conducive for radiation (because of the possibility to exchange genes) rather than preventing it (as would be expected from ecological theory).

Overall, I find this integrative study very well conducted, the methods are sound, and the results are of broad interest. I do have a few points, though, which would need further clarifications or which the authors should consider to incorporate in a revised version of their work:

>> We are pleased that the reviewer finds our manuscript interesting, well conducted and of broad interest. We would like to thank the reviewer for the helpful comments.

2.1. (1) I do think that the authors should not dismiss the differences between the two lakes in their consideration of whether or not an adaptive radiation takes place (that is, the abiotic factors). Lake Mweru is certainly larger, probably more stable over geological times, it is deeper (so there is real deep-water/open water niche), and it is situated at a lower altitude (these are all factors that contribute to adaptive radiation according to previous work by the same group; see Wagner et al.). Bangweulu, on the other hand, is a swamp after all. While I don't think that these abiotic factors are crucial for the story, these should be discussed in more detail.

>> We have now provided more details on the lakes and the differences between them in abiotic conditions (see also our reply to comment 1.2). Lake Bangweulu is not a swamp but a clear-water lake surrounded by extensive swamps. However, the swamp is often referred to under the same name, Bangweulu swamps, which can cause confusion. The cichlid species are shared between Lake Bangweulu and the Bangweulu swamps (and other smaller lakes inside the swamps). We have now added more details on the lake dimensions into the main text (see also our reply to comment 1.2). Even though the maximum lake depth does differ (10 vs 27 m), the mean depth is quite similar between the lakes because the deep parts of Lake Mweru are limited to a very small area (Bos, et al. 2006). Many other African lakes of depth similar or shallower than Bangweulu host cichlid radiations, and all of these are much smaller in terms of surface area (Wagner et al. 2012). Some of the better known examples include: Lake Ejagham 18 m max depth, surface 0.5 km², 2 radiations of 2 and 5 species; Lake Bermin, 14 m max depth, surface 0.6 km², a radiation of 9 species; Lake Saka, 4 m (except in a tiny spot of 12 m), a radiation of 2 or 3 species; Lake Natron, 1 m max depth, surface 759 m², a radiation of 5 species).

Nonetheless, in the revised version of the manuscript, we focus less on the comparison between the lakes and more on the coincidence of hybridization and ecological opportunity in Lake Mweru. Adaptive radiations are absent in Lake Bangweulu where hybridization is missing and in rivers where ecological opportunity is missing.

2.2. (2) Although not particularly important for the general interpretation of the results, I did not get the point why exactly the four age calibrations presented in Table S1 were chosen, as these do not include the minimum possible, nor the most up-to-date or the ones based on the largest data-sets. I would suggest to have a look at Michael Matschiner's review on that matter (Hydrobiologia cichlid special issue). Wouldn't it be possible to also try a demographic estimate of diversification times?

>> We agree that the review by Michael Matschiner published after submission of our manuscript is relevant. We thus cite it in the methods: "As the correct calibration of cichlid trees is controversial (Matschiner 2019; Schedel, et al. 2019), we used four different calibration sets with priors on the age of the Lake Malawi cichlids and on the modern haplochromines (see Table S1)." (L. 306)

Before that review, the most up-to-date calibration was the one by Irisarri et al. 2018 which includes the new cichlid fossil *Tugenchromis pickfordi*. To illustrate the uncertainty of the age estimates, we complemented this most up-to-date calibration set with classical calibration sets. As shown by Matschiner (2019), the Gondwana fragmentation calibration by Genner et al. (2007) used by us leads to the oldest divergence time estimates. We have also included the Friedman et al. (2013) calibration which Michael Matschiner reports as the second most recent calibration. We had not included the Near et al. (2013) calibration which provides the most recent time estimates but differs strongly from all

other calibration sets and is also not recommended by Matschiner (2019). We decided to keep the Gondwanan calibration even though Matschiner (2019) speaks against it, because all other scenarios cannot explain how cichlids got from African through India, Madagascar and the Neotropics in exactly the sequence that mirrors the relative timing of Gondwanan vicariance AND each of which happened prior to phylogenetic diversification within either of the continents. It is unlikely that transoceanic dispersal can explain why African cichlids are monophyletic to the exclusion of Neotropical cichlids. We thus do not want to imply that we know the final answer to the question of the correct calibration and include the entire breath of possible calibrations from other studies that use different lines of evidence.

2.3. (3) As mentioned above, I am generally quite happy with the methods. Nevertheless, I thought that the data-set at hand would ideally be run through Milan Malinsky et al.'s fineRADstructure software, which is specifically tailored for the RAD data such as the ones presented in this study. Such an analysis would allow for a detailed view on the co-ancestry of the taxa.

>> FineRADstructure is a great tool for detecting recent gene flow. Haplotypes that were shared through hybridization events a million years ago are expected to be heavily fragmented due to recombination and may not leave such a strong signal anymore. In addition, just like fineSTRUCTURE, fineRADstructure aims to find the closest relative for each haplotype and in the case of a radiation of hybrid origin, the closest relative is expected to be found in another member of the same radiation. However, as Malinsky et al. (2018) write in their last sentence, “the entries of the coancestry matrix provide an independent source of information on recent sample relatedness, complementing the information about older historical relationships; this may be especially informative in cases of recent gene-flow between populations.” We have thus run fineRADstructure to test for evidence of recent gene flow and potentially also weak evidence for ancestral hybridization. FineRADstructure correctly clusters the individuals of each radiation and species in Lake Mweru and supports the absence of ongoing gene flow as the members of each radiation show similar levels of haplotype sharing with members of all other lineages. There is one exception: One small-tooth serranochromine individual (“78322 Se. sp. “deep-red”) shows increased haplotype sharing with one large-tooth serranochromine individual (78584 Se. sp. “checkerboard”) potentially indicating gene flow a few generations ago. It is unclear which of the two individuals may be introgressed as the coancestry signal is symmetrical and neither of these individuals shows reduced haplotype sharing with members of their respective radiation. FineRADstructure supports a recent hybrid origin of “New Lufubu” as a mix between *Orthochromis* and *Pseudocrenilabrus* and it also supports the ancestral hybridization events we inferred with D statistics with weakly increased haplotype sharing between the radiations and parental lineage proxies.

We have now included the fineRADstructure results in the main text and show the results as new Supplementary Figures 6 and 7. We also provide all the source files for these and other figures.

2.4. (4) I feel that the authors should go over the taxonomy once more and provide updates, where necessary. For example, I believe that the taxon labelled as “New Lufubu” is in fact *O. indermauri* (Schedel et al. 2018); and perhaps some other taxa have been described (I might be wrong, though, but was not able to trace this back further).

>> Two of our included taxa have indeed recently been described by Schedel et al. (2018) and we use these new names in our manuscript (*O. mporokoso* and *O. katumbii*). However, we had already updated our taxon names for them and *O. indermauri* is not among them. *O. indermauri* does not occur in our study area but is confined to the lower Lufubu in the Lake Tanganyika lowlands, which we did not sample. "New Lufubu" is confined to the upper Lufubu River. These two taxa are not closely related and indeed upper and lower Lufubu are biogeographically distinct regions that have been isolated probably for millions of years. Both species are shown together in the mitochondrial phylogeny by Schedel, Musilova and Schliewen (2019) published in BMC Evolutionary Biology.

In addition, I have some rather minor comments/suggestions:

2.5. - 95: I don't want to advertise work on which I am co-author, but I think that Britta Meyer's PhD work (the Systematic Biology paper) is relevant here.

>> We now cite your paper alongside the others. (L. 96)

2.6. - 108/109: Out of interest: How do we know that *Tylochromis* is a strong swimmer compared to other cichlids? Is there a reference for this statement?

>> We are not aware of an experimental test of swimming ability. However, *Tylochromis* grows larger than any of the haplochromine species and is indeed a very good swimmer with a large caudal fin powered by a deep caudal peduncle. Many of its species occur in large rivers, such as the main stem of the Congo, and have wide distributions in these riverine landscapes, very different from haplochromines.

2.7. - 239: there's a double space before "paleolake"

>> Yes, it is now corrected.

2.8. - 279: To be fair, I think you should cite the original papers from where this number comes from, the Nagel paper (200k) and the Verheyen paper, which first reported this exact number (again, I am a co-author on the latter)

>> Verheyen et al. (2003) report a haplotype divergence age estimate for the LVRS of 98,000 to 132,700 years. In Nagel et al., (2000), the authors report that Lake Victoria cichlids are 100,000-200,000 years old. We have now changed ~150,000 to 100,000-200,000 and now also cite the two suggested papers which provide mitochondrial age estimates, alongside Bezault et al. (2011) which adds AFLP analyses and reviews lake ages as additional support for this age estimate (L. 311).

2.9. - 310: Chilwa or Chila? I think this makes a big difference (see also our latest work on sex chromosome evolution in *P. philander*...)

>> Lake Chilwa is correct. We had also included a sample of Lake Chila from you (SRR7185096) into the RAD tree and that one grouped with *P. philander* of Kafue River. Your paper on sex chromosome evolution in *P. philander* was interesting but we feel that it is not relevant here.

2.10. - 476, etc.: very good point.

>> thanks.

2.11. - 646: AFLP's: I see the point that you don't have the extracts anymore, but is it really necessary to report these outdated results and methods?

>> We have now moved the AFLP analyses to the Supplementary Notes. We think that it is important to have checked that none of these samples were misidentified but we agree that these AFLP results are not crucial to the main text.

2.12. - It would be great if you could improve a bit the .pdf of the detailed nuclear versus mitochondrial tree. Node labels, symbols and especially the branch support values are really difficult to read.

>> We have modified the pdf to allow for better reading of the bootstrap support labels by making them green instead of grey and slightly larger. Unfortunately, we cannot increase the font size of the labels as they would otherwise overlap. By providing the trees in pdf format as Supplementary Data, the readers should be able to zoom in for easier reading. We now also highlight the mitochondrially mismatched samples with bold green font in the mitochondrial tree.

I very much hope that my comments are useful.

With best wishes,

Walter

Yes, they were very useful. Thank you.

Reviewer #3:

I read carefully the manuscript entitled "The coincidence of ecological opportunity with hybridization explains the prevalence of rapid adaptive radiation". This study showed that adaptive radiations of four haplochromine lineages from Lake Mweru experienced hybridization in ancestral species with other cichlid lineages, but non-radiated haplochromine lineages from Lake Bangweulu haven't experienced hybridization. Ecological opportunity for adaptive radiation may not be different so much between the lakes; therefore, not only ecological opportunity but also hybridization is likely to be a key factor facilitating adaptive radiation. I didn't find problems in analyses and conclusion but very minor points regarding explanation.

>> We would like to thank the reviewer for the careful reading and the comments that have helped us to improve readability.

3.1. Line 151-154: This study called Pseudocrenilabrus radiation, Sargochromis radiation, large-tooth serranochromines, and small-tooth serranochromines "lacustrine radiations" of Lake Mweru. This information is very important in this study, but I didn't understand which are lacustrine radiations when I read the text first. Showing these radiations in this sentence, or marking these radiations with "L" or some character in Figure 1b will be helpful for readers.

>> Thank you for this suggestion. We have marked lacustrine radiations with (L) in Fig. 1b and also now mention this in that sentence: "Genomic and morphological data revealed that they evolved in four larger radiations confined to Lake Mweru (hereafter called lacustrine radiations, marked with (L) in Fig. 1b)" (L. 168). Similarly, in Fig. S4, we now more clearly highlight the lacustrine radiations with an (L) and by showing Lake Mweru taxa in red and taxa in adjacent rivers in orange. Also in Fig. 3, we now show riverine taxa in orange and the Lake Mweru taxa in red.

3.2. Line 192: It is difficult to follow this section. For example, I confused "Orthochromis species (line 198)" with orthochromines species at first. In the previous section, eight radiations were defined. In this section, explanation for each radiation may be easier to understand.

>> We have replaced the genus names by the names of the radiations in that section or added species names when reporting different ecologies for members of the same radiation.

References

- Bos AR, Kapasa CK, van Zwieten PAM 2006. Update on the bathymetry of lake mweru (zambia), with notes on water level fluctuations. African Journal of Aquatic Science 31: 145-150.
- Bos AR, Ticheler H 1996. A limnological update of the bangweulu fishery, zambia. DoF/BF/1996/Report no.26.
- Herdendorf CE 1982. Large lakes of the world. Journal of Great Lakes Research 8: 379-412.
- Hughes RH. 1992. A directory of african wetlands: IUCN.
- Marques DA, Meier JI, Seehausen O 2019. A combinatorial view on speciation and adaptive radiation. Trends Ecol Evol.
- Matschiner M 2019. Gondwanan vicariance or trans-atlantic dispersal of cichlid fishes: A review of the molecular evidence. Hydrobiologia 832: 9-37.
- Schedel FDB, Musilova Z, Schliewen UK 2019. East african cichlid lineages (teleostei: Cichlidae) might be older than their ancient host lakes: New divergence estimates for the east african cichlid radiation. BMC Evolutionary Biology 19: 94.
- Seehausen O 2004. Hybridization and adaptive radiation. Trends Ecol Evol 19: 198-207.
- Shiklomanov IA, Rodda JC. 2004. World water resources at the beginning of the twenty-first century: Cambridge University Press.
- Wagner CE, Harmon LJ, Seehausen O 2012. Ecological opportunity and sexual selection together predict adaptive radiation. Nature 487: 366-369.

Reviewers' Comments:

Reviewer #1:

Remarks to the Author:

I have very much enjoyed reading this paper and think that it will make an excellent and well cited contribution. The authors have done a great job at addressing the comments from the previous version. There are still some areas where I think that clarification could strengthen the arguments. The main issue is still just to address the confounding explanatory factors and in particular to clarify the role of ecological opportunity in facilitating adaptive radiation. Opportunity includes both size (extent or magnitude) of the lake that is being exploited (i.e., depth, area, or other factors relevant to the taxon) and the isolation. The authors focus on isolation, and the major environmental differences between lakes are mentioned, yet still not entirely integrated into the arguments. I think it would help clarify (and certainly not detract from) the arguments if the authors could clearly state that yes, the lake where they radiated (Mweru) is larger by all measurements and the larger size (depth) may (from what I understand of Wagner et al) be a key explanatory variable for radiation in Mweru and not Bangweulu.

Abstract

Depth is a critical explanatory factor that will be tested but is not mentioned. Maybe emphasize both depth (serving in the same way as area, presumably) and isolation. Otherwise it sounds like you have already made up your mind that hybridization is the answer.

Line 53 Maybe " Here, we use haplochromine cichlid assemblages in two previously unstudied African Great Lakes"

Line 57 So the greater biotic isolation might predict more ecological opportunity in Bangweulu, but the reduced depth would predict less, right?

Line 91 Co-occurrence of closely related lineages may also facilitate adaptive radiation by excluding secondary colonizers from their ancestral niche.

Line 97 It would help to clarify what this paragraph is setting up. So there are two largely unknown lakes, Mweru and Bangweulu. I would then point out how they differ in terms of the factors that provide ecological opportunity – depth and isolation. Then you can first talk about depth. Then about isolation.

Line 101 Lake Mweru is larger than Lake Bangweulu. Maybe add "in both area and depth".

Line 108 So depth is a better predictor of radiation than surface area. But then "radiations are known from lakes much smaller than Lake Bangweulu". Presumably smaller in surface area rather than depth (otherwise the phrasing doesn't follow). Just needs clarification

Line 110 I was then a bit lost as to the parameters of Lake Bangweulu that permitted in situ adaptive radiation in lakes elsewhere. Presumably this means that adaptive radiation occurs in other lakes of the same depth, given that area is not as good a predictor of whether a radiation occurs?

Line 112. You are now talking about isolation. It might help to make this a separate paragraph.

Line 116. I would put the aspects of size together, and keep the aspects of isolation and connection together.

Line 116 It would be great to know a bit more about the paleo history of the lakes at the time when the fish would have been diversifying.

Line 122 The fish fauna of Lake Mweru is composed of Congolese and Zambezian lineages ... Does this refer to haplochromines only? Just need to clarify relative to the next sentence where it says that Lake Bangweulu hosts one Congolese non-haplochromine cichlid

Line 124. Take out one "large".

Lines 126-128 Maybe clarify the caveats, otherwise it is somewhat confusing as to what is being tested, especially as we are told earlier that ecological opportunity requires both isolation and depth. So when you say that the study tests predictions of the isolation and the hybridization hypotheses, it just seems that the depth issue has been swept under the rug.

Line 440. Again, keep in mind that ecological opportunity includes space as well (you don't find adaptive radiations on remote atolls).

Line 483 I'm a bit lost by the statement that the smaller size of Lake Bangweulu is expected to result in lower species richness but cannot explain the absence of radiations. Remember that size is "the relative extent of something; a thing's overall dimensions or magnitude". And we know (from Wagner et al) that one aspect of size (depth) is most consistently associated with radiation, while another aspect of size (surface area) is not. However, Mweru is larger in both surface area and (in particular) depth, right? (At least from what is stated on L 101).

Line 486. Again, would be good to just make this really clear. Given that depth is the major predictor, does this mean that many lakes that are shallower than Lake Bangweulu host adaptive radiations?

Line 521 "respectively" implies that lack of hybridization is more important than lack of ecological opportunity. Not clear what this is based on?

Reviewer #2:

Remarks to the Author:

The authors have addressed my previous concerns. I thus think that the paper should be accepted for publication, and I would like to congratulate the authors to their work.

Point-by-point response

Reviewer #1:

1) I have very much enjoyed reading this paper and think that it will make an excellent and well cited contribution. The authors have done a great job at addressing the comments from the previous version. There are still some areas where I think that clarification could strengthen the arguments. The main issue is still just to address the confounding explanatory factors and in particular to clarify the role of ecological opportunity in facilitating adaptive radiation. Opportunity includes both size (extent or magnitude) of the lake that is being exploited (i.e., depth, area, or other factors relevant to the taxon) and the isolation. The authors focus on isolation, and the major environmental differences between lakes are mentioned, yet still not entirely integrated into the arguments. I think it would help clarify (and certainly not detract from) the arguments if the authors could clearly state that yes, the lake where they radiated (Mweru) is larger by all measurements and the larger size (depth) may (from what I understand of Wagner et al) be a key explanatory variable for radiation in Mweru and not Bangweulu.

>> As the reviewer has correctly pointed out, lake depth has been shown by some of us (CW and OS) to be a predictive factor of adaptive radiation. However, the difference in depth is restricted to a very small part of Lake Mweru (see map below) and even if one ignored this, the difference in depth is not sufficiently large to explain presence of adaptive radiations in 5 lineages in Lake Mweru and absence of adaptive radiation in all of the lineages that colonized Lake Bangweulu. Lake depth does predict adaptive radiation in cichlids in general, but the effect of lake depth as a predictor of adaptive radiation emerges strongly only at a scale of depth variation much larger than the difference in depth between our two lakes and we do observe radiation even in lakes much smaller and shallower than Bangweulu. For example, Lake Natron is only 1 m deep with a surface area of 759 m² and hosts a radiation of 5 species. Lake Bangweulu is thus clearly not too shallow for adaptive radiation to occur. We have now added a new Supplementary Figure 1 which is also shown below to illustrate this. The predictive power of lake depth on the presence of adaptive radiation (indicated as a speciation fraction >0) is strong when very deep lakes are included. All lakes of 100 m depth and deeper host radiations, but among the lakes shallower than 100 m, some host adaptive radiations and others do not.

In terms of area, Lake Bangweulu used to be much larger until it started draining into the Luapula (Cotterill & de Witt, 2011, S Afr J Geol). In addition, even today most fish of Lake Bangweulu also occur both in the vast swamps around the lake. If one counted the swamps to the total size of the habitat, Lake Bangweulu would be larger in area than Lake Mweru. In addition, the presence of a greater number of fish species of other families in Lake Mweru could also more strongly restrict the ecological opportunity of haplochromine cichlids in Lake Mweru than in Lake Bangweulu.

We thank the reviewer for highlighting that we had previously not been clear enough on this topic and pointing us to parts of the manuscript where this could be addressed more clearly.

Bathymetric map of Lake Mweru from (Bos & Zwieten., 2006). The part of Lake Mweru that is deeper than Lake Bangweulu (>10 m) is highlighted in yellow.

Supplementary Fig. 1: Lake depth does not explain the lack of speciation in Lake Bangweulu and the presence of 5 adaptive radiations in Lake Mweru. Blue and black dots depict lakes with and without adaptive radiation, respectively, whereas orange dots highlight Lakes Mweru and Bangweulu. The fraction of species that evolved through *in situ* speciation (speciation fraction) is not significantly correlated with lake depth. Lakes similar in depth as Lakes Bangweulu and Mweru can host cichlid communities that are fully assembled through colonization, completely derived from *in situ* speciation, or anything in between. Figure adapted from Wagner et al., 2014, Ecology Letters.

2) Abstract

Depth is a critical explanatory factor that will be tested but is not mentioned. Maybe emphasize both depth (serving in the same way as area, presumably) and isolation. Otherwise it sounds like you have already made up your mind that hybridization is the answer.

>> As outlined above, we do not think that the difference in depth between Lakes Mweru and Bangweulu should strongly affect the propensity of adaptive radiation in either. The scale of variation in lake depth at which it emerges as a strong predictor of adaptive radiation by far exceeds the difference between these two lakes. Both lakes are deep enough for adaptive radiation to be possible but not deep enough to be in the depth range where all lakes host adaptive radiation (>100 m). Area does not predict the propensity of adaptive radiation of cichlid fishes in African lakes (Wagner et al., 2014). The biggest difference between Lakes Mweru and Bangweulu is the drainage system history influencing the number of species (cichlids and other fish) that colonized either lake, and hence both opportunity for competition and for hybridization. We now write on L39: “Greater biotic isolation (fewer lineages) predicts fewer constraints by competition and hence more ecological opportunity in Lake Bangweulu, whereas opportunity for hybridization predicts increased genetic potential in Lake Mweru.”

3) Line 53 Maybe “Here, we use haplochromine cichlid assemblages in two previously unstudied African Great Lakes”

>> This comment does not apply anymore to the shortened abstract as we now do not mention anymore that the lakes were previously unstudied.

4) Line 57 So the greater biotic isolation might predict more ecological opportunity in Bangweulu, but the reduced depth would predict less, right?

>> Please see our reply to comment 1.

5) Line 91 Co-occurrence of closely related lineages may also facilitate adaptive radiation by excluding secondary colonizers from their ancestral niche.

>> This is an interesting point. However, whereas this would lead to ecological character displacement facilitating the coexistence of the colonizing lineages, it would be unlikely to increase the number of species above the number of colonizing lineages. Here, we discuss examples of interactions among cichlid lineages and as far as we know, in cichlids lacustrine adaptive radiations are generally *in situ* radiations. We do not know of examples where adaptive radiation proceeds through rounds of distinct colonization events, followed by character displacement upon secondary sympatry (i.e. the Darwin’s finch scenario).

6) Line 97 It would help to clarify what this paragraph is setting up. So there are two largely unknown lakes, Mweru and Bangweulu. I would then point out how they differ in terms of the factors that provide ecological opportunity – depth and isolation. Then you can first talk about depth. Then about isolation.

>> We have now split up this paragraph into one paragraph about lake size and one paragraph about geographic isolation.

7) Line 101 Lake Mweru is larger than Lake Bangweulu. Maybe add “in both area and depth”.

>> We now write on L. 83 “The two lakes differ somewhat in their dimensions.” and provide a detailed comparison of lake depth and area and their expected effects on the propensity of adaptive radiation in the rest of the paragraph starting on L84:

“Lake Mweru has a surface area of 5,100 km²²⁹ with a mean depth of 7.5 m and a maximum depth of 27 m³⁰. The surface area of Lake Bangweulu is subject to strong seasonal changes and is reported as 2,500-5,000 km²^{29,31,32}, with a mean depth of 4.7 m and a maximum depth of 10.4 m³³ (Supplementary Note 1). Previous work shows that for African lakes in general, lake depth is a predictive factor for adaptive radiation in cichlids, along with sexual dichromatism and energy measured as solar radiation¹². However, the predictive power of lake depth on adaptive radiation strongly emerges only when lakes deeper than 100 m are included in the analysis, all of which host adaptive radiations. The scale at which depth is predictive of adaptive radiation vastly exceeds the difference in depth between Lakes Mweru and Bangweulu. Of the lakes with depths similar to Lakes Mweru and Bangweulu, some host cichlid communities, whereas others show no in situ speciation at all (Supplementary Fig. 1). Lake surface area, on the other hand, predicts the species richness of cichlid assemblages but does not predict whether or not a radiation occurs¹². Radiations are equally likely to occur in lakes with small surface area but they attain lower species richness than large lakes^{12,18}. The slightly smaller surface area of Lake Bangweulu would make us expect slightly lower species richness but is not expected to constrain the occurrence of adaptive radiation.”

8) Line 108 So depth is a better predictor of radiation than surface area. But then “radiations are known from lakes much smaller than Lake Bangweulu”. Presumably smaller in surface area rather than depth (otherwise the phrasing doesn’t follow). Just needs clarification

>> Thanks for pointing this out. Radiations are known from lakes that are both smaller in depth and in area than Lake Bangweulu. We show the relationship with depth in the new supplementary figure (and our reply to comment 1). We have reformulated this paragraph and removed that sentence (see our reply to comment 7).

9) Line 110 I was then a bit lost as to the parameters of Lake Bangweulu that permitted in situ adaptive radiation in lakes elsewhere. Presumably this means that adaptive radiation occurs in other lakes of the same depth, given that area is not as good a predictor of whether a radiation occurs?

>> Yes, some lakes shallower than Lake Bangweulu host adaptive radiations. The scale at which variation in lake depth strongly influences the propensity of adaptive radiation by far exceeds the difference in depth between Lake Mweru and Bangweulu. In addition, only maximum lake depth is considered here, but the maximum depth of Lake Mweru is restricted to a very small part of the lake (see the map in our reply to comment 1).

10) Line 112. You are now talking about isolation. It might help to make this a separate paragraph.

>> Yes, we have done that. Thank you for this suggestion.

11) Line 116. I would put the aspects of size together, and keep the aspects of isolation and connection together.

>> This sentence is about the age of the lakes, not about size. We have reformulated it to make this clearer.

12) Line 116 It would be great to know a bit more about the paleo history of the lakes at the time when the fish would have been diversifying.

>> Yes, we agree that this would be great. We have now extended the information in Supplementary Note 1 a bit but not much more is known, unfortunately.

13) Line 122 The fish fauna of Lake Mweru is composed of Congolese and Zambezian lineages ... Does this refer to haplochromines only? Just need to clarify relative to the next sentence where it says that Lake Bangweulu hosts one Congolese non-haplochromine cichlid

>> This applies to the entire fish fauna. The rich fish diversity of Lake Mweru is a faunal mix of Congolese and Zambezian lineages in haplochromines, in non-haplochromine cichlids (see Fig. 1) and also in other fish families. We have now hopefully made this clearer by splitting the sentence into two: L. 112

“Indeed, the fish fauna of Lake Mweru is composed of Congolese and Zambezian lineages, whereas Lake Bangweulu hosts almost exclusively Zambezian lineages. The only cichlid exception is *Tylochromis bangwelensis*, a large Congolese non-haplochromine cichlid with strong swimming capacities that seems to have colonized Lake Bangweulu upstream from Lake Mweru.”

14) Line 124. Take out one “large”.

>> Thank you for finding this mistake. We have corrected it.

15) Lines 126-128 Maybe clarify the caveats, otherwise it is somewhat confusing as to what is being tested, especially as we are told earlier that ecological opportunity requires both isolation and depth. So when you say that the study tests predictions of the isolation and the hybridization hypotheses, it just seems that the depth issue has been swept under the rug.

>> We think that the difference in depth is small and unlikely to strongly affect ecological opportunity, whereas the difference in isolation is large, such that one lake is a site of major faunal interchange between two of Africa’s biggest river systems, whereas the other lake is entirely retained within just one of these. We hope that this is now clearer in the introduction and we do discuss the caveats of the potential influence of depth later in the manuscript.

16) Line 440. Again, keep in mind that ecological opportunity includes space as well (you don’t find adaptive radiations on remote atolls).

>> Yes, agreed. However, in this sentence we discuss the effect of isolation on ecological opportunity.

17) Line 483 I'm a bit lost by the statement that the smaller size of Lake Bangweulu is expected to result in lower species richness but cannot explain the absence of radiations. Remember that size is "the relative extent of something; a thing's overall dimensions or magnitude". And we know (from Wagner et al) that one aspect of size (depth) is most consistently associated with radiation, while another aspect of size (surface area) is not. However, Mweru is larger in both surface area and (in particular) depth, right? (At least from what is stated on L 101).

>> We have now reformulated this section to make it clearer. We now state on L. 413: "The smaller area of Lake Bangweulu compared to Lake Mweru is expected to result in lower species richness in Lake Bangweulu, but cannot explain the complete absence of any *in situ* speciation because lake area does not predict whether or not cichlid fish adaptive radiations happen. Lake depth does predict adaptive radiation in cichlids in general, but the effect of lake depth as a predictor of adaptive radiation emerges strongly only at a scale of depth variation much larger than the difference in depth between our two lakes and we do observe radiation even in lakes much smaller and shallower than Lake Bangweulu¹²."

18) Line 486. Again, would be good to just make this really clear. Given that depth is the major predictor, does this mean that many lakes that are shallower than Lake Bangweulu host adaptive radiations?

>> Yes. See our reply to comment 17.

19) Line 521 "respectively" implies that lack of hybridization is more important than lack of ecological opportunity. Not clear what this is based on?

>> This is not what we wanted to imply. This sentence stated that the absence of adaptive radiation in Lake Bangweulu is likely explained by lack of hybridization and the absence of adaptive radiation in the rivers is likely explained by the lack of ecological opportunity (despite that hybridization occurred). Yes, the ecological opportunity in Lake Bangweulu is enough for adaptive radiation given that other lakes that are smaller in depth and area do harbor adaptive radiations. We have now split this sentence into two to make it clearer. We now write on L. 457: "The absence of radiations in Lake Bangweulu may be explained by lack of hybridization (despite ecological opportunity). The absence of radiations in the rivers around Lake Mweru may be explained by the lack of ecological opportunity (despite hybridization). This is consistent with coincidence of ecological opportunity and genetic opportunity for hybridization being required for rapid adaptive radiation (besides the other prerequisites that are given in haplochromines as explained above)."

Reviewer #2:

The authors have addressed my previous concerns. I thus think that the paper should be accepted for publication, and I would like to congratulate the authors to their work.

>> We are glad that the reviewer is happy with our revised version and would like to thank him again for the helpful comments.